Resource

# Development of a new monoclonal antibody specific to mouse Vγ6 chain

Shinya Hatano[1] ⬤, Xin Tun[1], Naoto Noguchi[1], Dan Yue[1,2], Hisakata Yamada[1], Xun Sun[2], Mitsuru Matsumoto[3], Yasunobu Yoshikai[1]

There are seven Vγ gene segments in the TCR γ chain loci of mice. We developed monoclonal antibodies (mAbs) specific to the Vγ6 chain (Heilig & Tonegawa nomenclature). By immunizing Vγ4/6 KO mice with complementarity-determining region peptides in Vγ6 chains, we generated three hybridomas. These hybridomas produced mAbs capable of cell surface staining of Vγ6/Vδ1 gene–transfected T-cell line lacking TCR as well as of Vγ1⁻ Vγ4⁻ Vγ5⁻ Vγ7⁻ γδ T cells and the CD3^high TCRδ^int γδ T cells in various organs. The location of Vγ6⁺ γδ T cells, which peaked in the newborn thymus, was associated with mTEC. In vivo administration of clone 1C10-1F7 mAb impaired protection against *Klebsiella pneumoniae* infection but ameliorated psoriasis-like dermatitis induced by imiquimod treatment. These new mAbs are useful to elucidate the development, location, and functions of Vγ6 γδ T cells in mice.

## Introduction

TCR γ chain loci have three functional Cγ genes (Cγ1, Cγ2, and Cγ4) and one nonfunctional pseudo Cγ gene (φCγ3), four joining segments, including one pseudogene (Jγ1, Jγ2, φJγ3, and Jγ4), and seven variable (Vγ) gene segments (Saito et al, 1984). The Vγ genes are Vγ1, Vγ2, Vγ3, Vγ4, Vγ5, Vγ6, and Vγ7, using the Heilig & Tonegawa nomenclature (Heilig & Tonegawa, 1986), which we used here, or Vγ1.1, Vγ1.2, Vγ1.3, Vγ2, Vγ3, Vγ4, and Vγ5, using the Garman nomenclature (Garman et al, 1986). Gene rearrangement of γδ TCR loci occurs at an early stage in the fetal thymus before αβ TCR genes rearrange in the thymus. Mouse fetal development is characterized by producing waves of γδ T-cell populations that use different Vγ chains (Chien et al, 1987; Ito et al, 1989). During embryonic development, the first T cells to appear from approximately embryonic day 12 (E12) to E16 carry γδ TCR composed of Vγ5 and Vδ1 chains (Vγ5Jγ1 and Vδ1Dδ2Jδ2), which populate the epidermis, and these T cells, which become wedged among keratinocytes and adopt a dendritic-like

form, are termed dendritic epidermal T cells (dETCs) (Asarnow et al, 1988; Havran et al, 1989; Havran & Allison, 1990). The second T cells appearing from E14 to birth carry Vγ6 paired with Vδ1 of γδ TCR (Vγ6Jγ1 and Vδ1Dδ2Jδ2), which home to the epithelia of the reproductive tract, tongue, lungs, peritoneal cavity (PEC), skin dermis, colon-lamina propria lymphocytes (c-LPLs) and adipose tissue as tissue-associated cells (Itohara et al, 1990; Mokuno et al, 2000; Roark et al, 2004; Cai et al, 2011; Sun et al, 2013; Kohlgruber et al, 2018). These two subsets bear truly invariant TCRs without junctional diversity, even no nucleotides in the TCR gene junction, and are essentially an oligoclonal population of cells. The following waves are Vγ4⁺ T cells from E16 onward and Vγ1⁺ T cells from E18 onward, all of which show junctional diversity in complementarity-determining region (CDR) 3. At the periphery, most of the spleen and LN γδ T cells express Vγ1 and Vγ4, whereas Vγ7-expressing γδ T cells are more prevalent in intestinal intraepithelial cells (i-IELs) (Goodman & Lefrancois, 1989). This bias in Vγ usage has led to the suggestion that Vγ-encoded residues enable these T cells to respond to Ag unique to their resident tissues. Recently, Vγ7⁺ i-IEL are reported to respond to epithelial butyrophilin-like (Btnl) protein of the B7 superfamily using germ line–encoded motifs distinct from CDRs within the Vγ7 chain (Di Marco Barros et al, 2016; Melandri et al, 2018). Thus, the bias of Vγ usage in various mucosal tissues has led to the suggestion that Vγ-encoded residues enable these T cells to respond to agonists unique to their resident tissues.

All monoclonal antibodies (mAbs) specific to Vγ chains, except for Vγ3 and Vγ6, are currently available for cell surface staining (Goodman & Lefrancois, 1989; Havran et al, 1989; Itohara et al, 1989, 1990; Dent et al, 1990; Goodman et al, 1992; Pereira et al, 1995; Mallick-Wood et al, 1998; Grigoriadou et al, 2002). We have detected Vγ6 γδ T cells indirectly by expressing Vγ6-encoding mRNA (Mokuno et al, 2000; Murakami et al, 2016). Roark et al reported that 17D1 mAb, which was first thought to detect dETCs bearing Vγ5/Vδ1 (Mallick-Wood et al, 1998), could also bind Vγ6/Vδ1 γδ T cells if their TCR was first complexed to an anti-Cδ mAb (GL3) (Roark et al, 2004). Furthermore, Paget et al (2015) identified IL-17A–producing Vγ6/Vδ1 γδ T cells as CD3^bright γδ T cells by anti-CD3ε mAb. However, detailed characteristics of Vγ6⁺ γδ T cells remain obscure because of the lack

[1]Division of Host Defense, Medical Institute of Bioregulation, Kyushu University, Fukuoka, Japan    [2]Department of Immunology, China Medical University, Shenyang, China    [3]Division of Molecular Immunology, Institute for Enzyme Research, Tokushima University, Tokushima, Japan

Correspondence: hatano@bioreg.kyushu-u.ac.jp

of Vγ6-specific mAb. In this study, we developed new mAbs specific to the murine Vγ6 chain and report the successful production, characterization, and in vitro effects of a novel anti-Vγ6 mAb with potential applications in elucidating roles of Vγ6 γδ T cells in infection and inflammation in mice.

# Results and Discussion

### Production of mAbs against Vγ6 TCR available for cell surface staining

The V part of an Ab, including the unique Ag-binding site, is known as the idiotype. When one Ab binds to an idiotype of another Ab, it is referred to as an anti-idiotypic Ab (Jerne, 1974). An anti-idiotypic network exists in autoimmune diseases, regulating the production of autoantibodies, or the idiotypic response (Menshikov et al, 2015). However, in a healthy state, because of tolerance to self Ag (Kappler et al, 1988), it is difficult to produce anti-idiotypic Abs against the V repertoire of conventional Abs. Similarly, it can be speculated that normal mice may be tolerant to the V region of the TCR including the Vγ6 chain. Therefore, we used Vγ4/6 KO mice (Sunaga et al, 1997) as recipients for the development of mAbs specific to the Vγ6 chain. The most likely immunogenic epitopes for staining the Vγ6 chain lie within the hypervariable CDR that provides most binding contacts. However, the CDR3 of the Vγ5 and Vγ6 chain of the invariants Vγ5Vδ1 and Vγ6Vδ1 are the same. So, we selected peptides from CDR1 or CDR2 as immunogens (WHO-IUIS, 1995). Two types of peptides were synthesized from truncated regions of CDR1 (Vγ6$_{21-35}$) and CDR2 (Vγ6$_{50-66}$) in Vγ6 chains (WHO-IUIS, 1995). Vγ4/6 KO mice were immunized s.c. with KLH-conjugated peptides emulsified with CFA. After a consecutive booster, iliac LNs were collected from immunized mice and were fused with SP2/0-Ag14 using polyethylene glycol (Yokoyama et al, 2010). The supernatant of the hybridoma culture was collected, and Ab titers were determined by ELISA on plates coated with BSA-conjugated CDR1 or CDR2. Among more than 800 hybridomas within each group, we selected 286 and 199 hybridomas that secreted mAbs specific to CDR1 (Vγ6$_{21-35}$) and CDR2 (Vγ6$_{50-66}$), respectively by ELISA.

To further select mAbs available for cell surface staining, we screened for those mAbs capable of staining TG40, a cell surface TCR–negative and intracytoplasmic CD3–positive mutant of the 21.2.2 mouse T-cell line (Sussman et al, 1988; Ohno et al, 1991), which was transfected with Vγ6/Vδ1 genes (Vγ6Vδ1-rCD2) or with Vγ5/Vδ1 genes (Vγ5Vδ1-rCD2). After screening, we selected three hybridomas; clones: 1C10-1F7, 2G2-2A3, and 5E10-C12 producing mAbs, which stained only TG40 cells transfected with Vγ6/Vδ1 genes but not TG40 cells transfected with Vγ5/Vδ1 genes (Fig 1A). Interestingly, all clones were derived from mice immunized with the Vγ6 CDR2 (Vγ6$_{50-66}$) peptide.

17D1 mAb, which detects dETC-bearing Vγ5/Vδ1 (Mallick-Wood et al, 1998), reportedly binds Vγ6/Vδ1 γδ T cells if its TCR is first complexed to an anti-Cδ mAb (clone: GL3) (Roark et al, 2004). This suggests that the conformational change by binding GL3 mAb may allow strong 17D1 mAb binding and that the Vγ6 chain may have a unique structural association with Cδ and the CD3ε complex. We

also found that TG40 cells transfected with the Vγ6/Vδ1 genes were stained with 17D1 mAb only if their TCR was first complexed to an anti-Cδ mAb (GL-3) (Roark et al, 2004). On the other hand, 1C10-1F7 mAb was capable of staining the cell line strongly without GL3 binding (Fig 1B). There are only a few murine γδ TCR structures currently known (Chien & Konigshofer, 2007). Our mAbs specific to Vγ6 chain may be useful for further analyses to elucidate the 3D structures of TCR Vγ6.

We next determined the nucleotide sequences and the reduced aa sequences of the H and L chains of Vγ6-specific mAbs of 1C10-1F7, 2G2-2A3, and 5E10-C12. The V-D-J genes of the H chain, the V-J genes of the L chain of these mAbs, and aa sequences of each CDR3 are shown in Fig 1C (Fig 1C). 1C10-1F7 and 5E10-C12 mAbs used the same VH, DH, VL, and JL genes, showing the same CDR3. 2G2-2A3 mAb used JH and JL genes different from those in 1C10-1F7 and 5E10-C12 mAbs showing different CDR3 sequences. The isotypes of the H and L chains are 1C10-1F7 (IgG1, κ), 2G2-2A3 (IgG1, κ), and 5E10-C12 (IgG2b, κ).

### Confirmation of specificity by immunofluorescence studies against γδ T cells in various tissues

Vγ6 γδ T cells are relatively abundant in the epithelia of the PEC, reproductive organs (vagina/uterine cervix), lungs, and c-LPL as tissue-associated cells (Itohara et al, 1990; Mokuno et al, 2000; Murakami et al, 2016; Sun et al, 2013). We stained for Vγ6+ γδ T cells in the nonlymphoid tissues from C57BL/6, BALB/c, and Vγ4/6 KO mice with 1C10-1F7, 2G2-2A3, or 5E10-C12 mAbs. Most of the Vγ1− Vγ4− Vγ5− γδ T cells in the reproductive organs and large proportions of the Vγ1− Vγ4− Vγ5− γδ T cells in the PEC and c-LPL from C57BL/6 were positively stained with 1C10-1F7 mAb (Figs 2A and S1A). 2G2-2A3 and 5E10-C12 mAbs also stained Vγ1− Vγ4− Vγ5− γδ T cells of these organs (Fig 2B). The Vγ6+ γδ T cells in the PEC, reproductive organs, and c-LPL from BALB/c mice were also stained with 1C10-1F7 mAb (Fig 2C). The Vγ6+ γδ T cells stained with 1C10-1F7, 2G2-2A3, or 5E10-C12 mAbs were absent in the PEC from Vγ4/6 KO mice (Fig 2D). We selected 1C10-1F7 mAb (IgG1, κ) with the highest affinity for further experiments. The Vγ1+ or Vγ4+ γδ T cells in the PEC were not stained with 1C10-1F7 mAb (Fig S1B). dETCs, which express Vγ5 exclusively (Asarnow et al, 1988; Havran & Allison, 1990; Havran et al, 1989) and i-IEL, which contains abundant levels of Vγ7+ γδ T cells (Goodman et al, 1992; Pereira et al, 1995), were not stained with 1C10-1F7 mAb (Fig S1C and D). To further confirm that 1C10-1F7 mAb recognizes the Vγ6+ chain on γδ T cells, we sorted 1C10-1F7+ γδ T cells from the PEC and examined the expression of TCR Vγ6 gene by RT-PCR and nucleotide sequences. The 1C10-1F7+ γδ T cells sorted from the PEC expressed Vγ6-specific transcripts, and all Vγ6-Jγ1 transcripts showed no junctional diversity (Fig S2). This resulted in in-frame invariant canonical sequences, which are preferentially expressed in Vγ6+ γδ T cells in the fetal thymus (Lafaille et al, 1989), reproductive organs (Itohara et al, 1990), and the PEC (Mokuno et al, 2000).

We next examined the location of Vγ6+ γδ T cells by immunohistochemical analysis with 1C10-1F7 mAb in the uterine cervix, in which most of γδ T cells were positively stained with 1C10-1F7 mAb. Consistent with a previous report (Itohara et al, 1990), the Vγ6+ γδ T

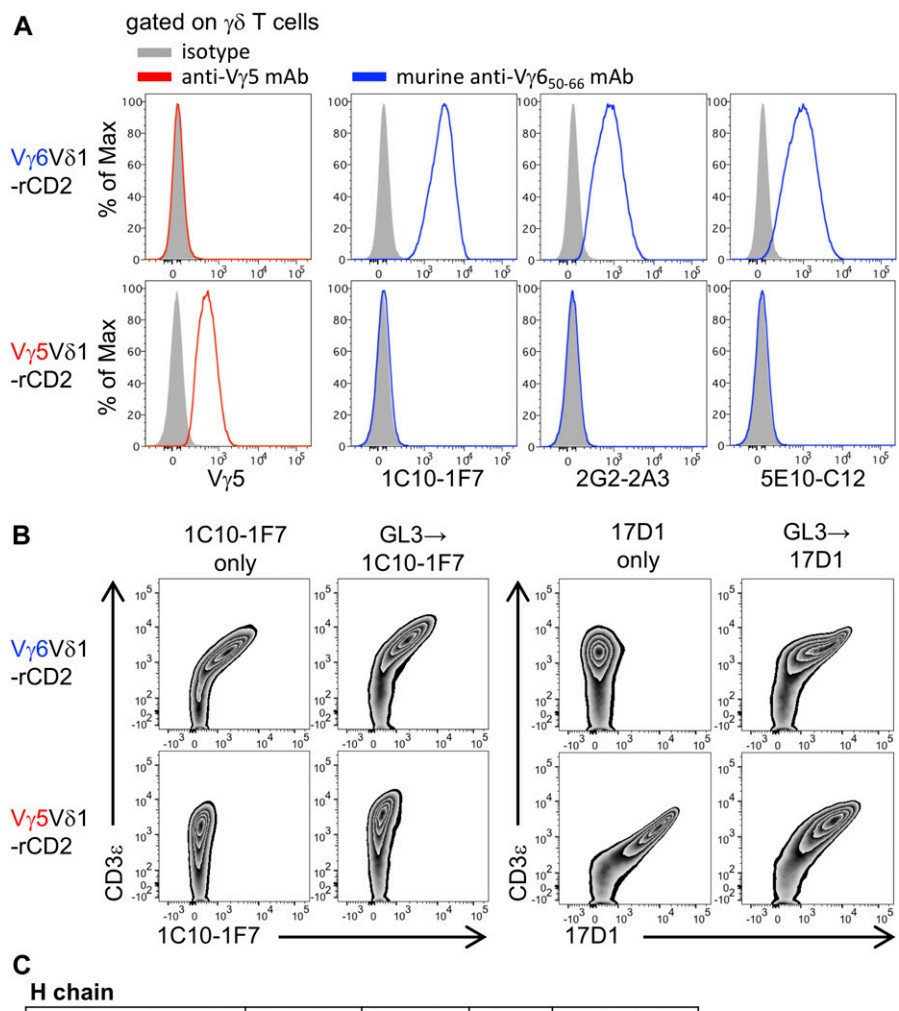

**Figure 1.  New mAbs are available for cell surface staining for Vγ6 TCR.**
**(A)** TG40 introduced with the Vγ6/Vδ1 gene (Vγ6Vδ1-rCD2) or the Vγ5/Vδ1 gene (Vγ5Vδ1-rCD2) were stained with mAbs from 1C10-1F7, 2G2-2A3, and 5E10-C12 or anti-Vγ5 mAbs. Histograms show expression of Vγ5 and Vγ6 on Vγ6Vδ1-rCD2 or Vγ5Vδ1-rCD2 after gating on TCRδ⁺CD3ε⁺. **(B)** Vγ6Vδ1-rCD2 and Vγ5Vδ1-rCD2 were stained with 1C10-1F7 or 17D1 with or without prestaining with GL3. The zebra plot shows 1C10-1F7 staining and 17D1 staining of Vγ6Vδ1-rCD2 and Vγ5Vδ1-rCD2, respectively. **(C)** V-D-J genes of the H chain, V-J genes of the L chain, and aa sequences of each CDR3 of 1C10-1F7, 2G2-2A3, and 5E10-C12 mAbs.

**H chain**

|  | V gene | D gene | J gene | CDR3 |
|---|---|---|---|---|
| 1C10-1F7 (IgG1,κ) | IGHV5-17 | IGHD4-1 | IGHJ2 | ATGTGFDS |
| 2G2-2A3 (IgG1,κ) | IGHV5-17 | IGHD4-1 | IGHJ3 | ASGTGFGC |
| 5E10-C12 (IgG2b,κ) | IGHV5-17 | IGHD4-1 | IGHJ2 | ATGTGFDS |

**L chain**

|  | V gene | J gene | CDR3 |
|---|---|---|---|
| 1C10-1F7 (IgG1,κ) | IGKV1-110 | IGKJ4 | SQSTHVPFT |
| 2G2-2A3 (IgG1,κ) | IGKV1-110 | IGKJ2 | SQSTHVPYT |
| 5E10-C12 (IgG2b,κ) | IGKV1-110 | IGKJ4 | SQSTHVPFT |

cells were abundantly present sub-epithelially, just under the cervical epithelium (Fig 2E).

### Ontogenic wave of Vγ6 γδ T cells

γδ T cells expressing Vγ5, Vγ6, Vγ4, Vγ1, and Vγ7 TCR develop sequentially in this order in the fetal thymus around E12 and E16 (Chien et al, 1987; Ito et al, 1989). We consistently found that Vγ5⁺ γδ T cells were abundant in the fetal thymus at earlier stages of development, and the percentage decreased from E16 onward during embryonic development. γδ T-cell waves from E16 onward were Vγ4⁺ cells. The number of Vγ6⁺ γδ T cells increased gradually during embryonic development, reaching a peak at neonatal stage from birth to day 3 (Fig 3A–C). These results are consistent with previous data showing a peak of Vγ6 γδ T cells at birth (Ito et al, 1989).

Reconstitution of lethally irradiated adult mice with BM or fetal liver (FL) resulted in failure to generate Vγ5⁺ γδ T cells, implying that the development of fetal type γδ T cells requires an embryonic thymus per se (Vantourout & Hayday, 2013; Cai et al, 2014). We also examined whether Vγ6⁺ γδ T cells in lethally irradiated mice reconstituted with BM or FL cells and Vγ6⁺ γδ T cells in the periphery. Vγ4⁺ γδ T cells were detected in the PEC of either of these reconstituted mice, but Vγ6⁺ γδ T cells were not (Fig S3A–C). These

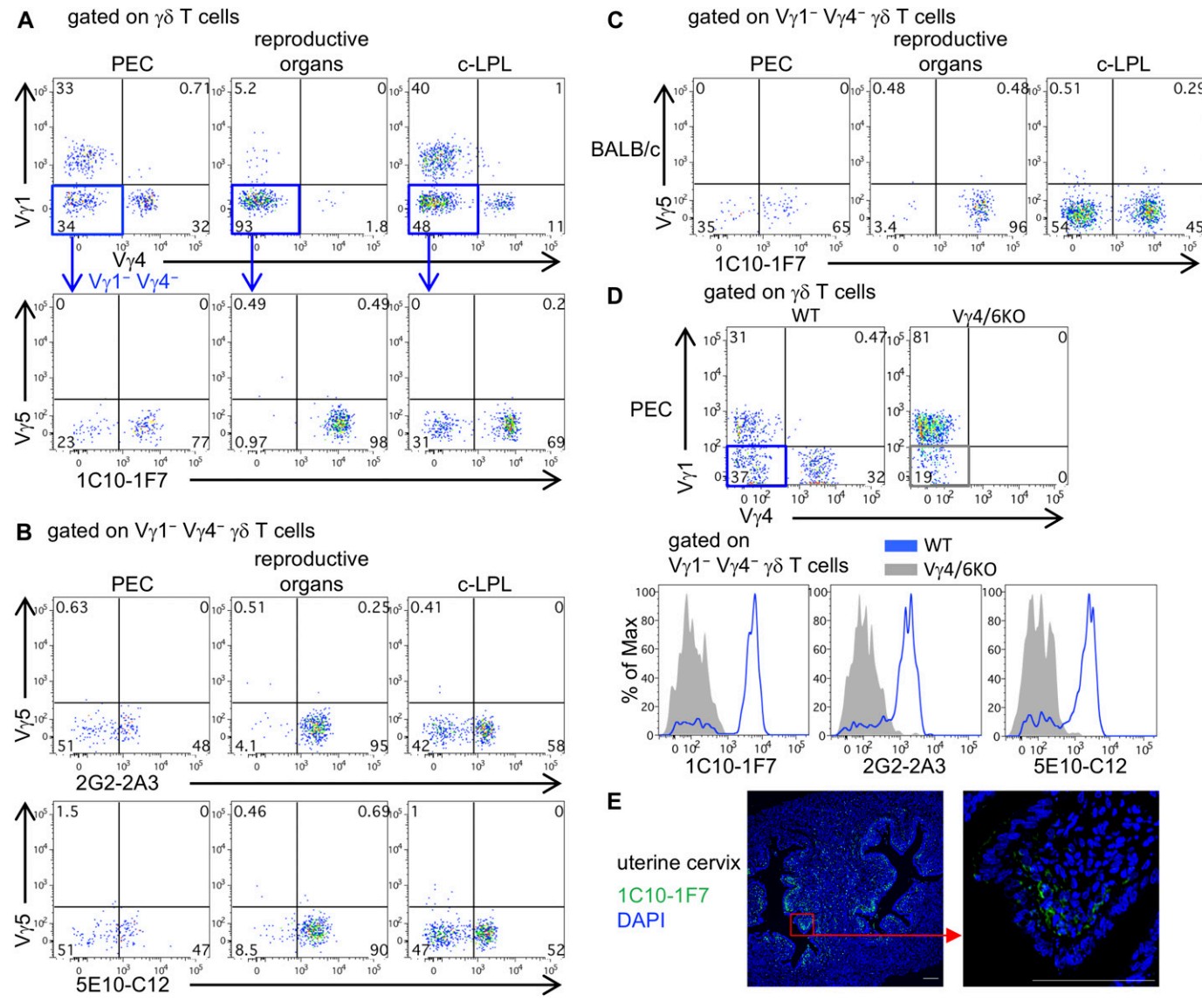

**Figure 2. New mAbs are useful for analyzing Vγ6 γδ T cells in various tissues.**
**(A)** 1C10-1F7 mAb staining in C57BL/6 mice. Representative upper dot plots are shown after gating γδ T cells and lower dot plots are shown after gating Vγ1⁻ Vγ4⁻ γδ T cells in the PEC, reproductive organs (vagina/uterine cervix), and c-LPL. **(B)** 2G2-2A3 or 5E10-C12 mAbs staining in C57BL/6 mice. Representative dot plots are shown after gating Vγ1⁻ Vγ4⁻ γδ T cells in indicated organs. **(C)** 1C10-1F7 mAb staining in BALB/c mice. Representative dot plots are shown after gating Vγ1⁻ Vγ4⁻ γδ T cells in indicated organs. **(D)** Dot plots are shown after gating γδ T cells in PEC from WT (C57BL/6) or Vγ4/6 KO mice. Histograms show the expression of 1C10-1F7, 2G2-2A3, or 5E10-C12 on Vγ1⁻ Vγ4⁻ γδ T cells of PEC from WT or Vγ4/6 KO mice. **(E)** Paraformaldehyde-fixed paraffin section of uterine cervix from WT mice was stained with Alexa Fluor 647–conjugated 1C10-1F7 mAb (green) and DAPI (blue). Right panel shows higher magnification image in the red square region of left panel. All scale bars represent 100 μm.

results suggest that Vγ6 γδ T cells are of the fetal type, and the development of γδ T cells may require an embryonic thymus per se.

We recently reported that IL-17–producing γδ T cells developed at the CD4⁻ CD8⁻ double-negative (DN)2b stage, which is located in the medulla (Shibata et al, 2014). Consistent with this report, immunohistochemical staining with 1C10-1F7 mAb revealed that Vγ6⁺ γδ T cells were located at the medulla of the neonatal thymus (Fig 3D). Double staining with 1C10-1F7 mAb and medullary thymic epithelial cell (mTEC)–specific mAb ER-TR5 (Van Vliet et al, 1984) suggested cross talk between Vγ6 and mTEC for selection (Fig 3E). It

has been reported that IL-17⁺ Vγ6⁺ Vδ1⁺ T cells are enriched in several organs of mice deficient in autoimmune regulator (Aire) gene, which is expressed by the mTEC (Fujikado et al, 2016). Nitta et al recently reported that IL-17⁺ Vγ6⁺ T cells were substantially enhanced in TN mice, which have no mature cortical TECs (cTECs) and substantially reduced number of mTECs in thymus (Nitta et al, 2015). Taken together, it is suggested that mTECs negatively regulate the development of IL-17⁺ Vγ6⁺ γδ T cells in the thymus. However, this is only speculation and further experiments need to clarify the significance of interaction of Vγ6⁺ γδ T cells and mTECs.

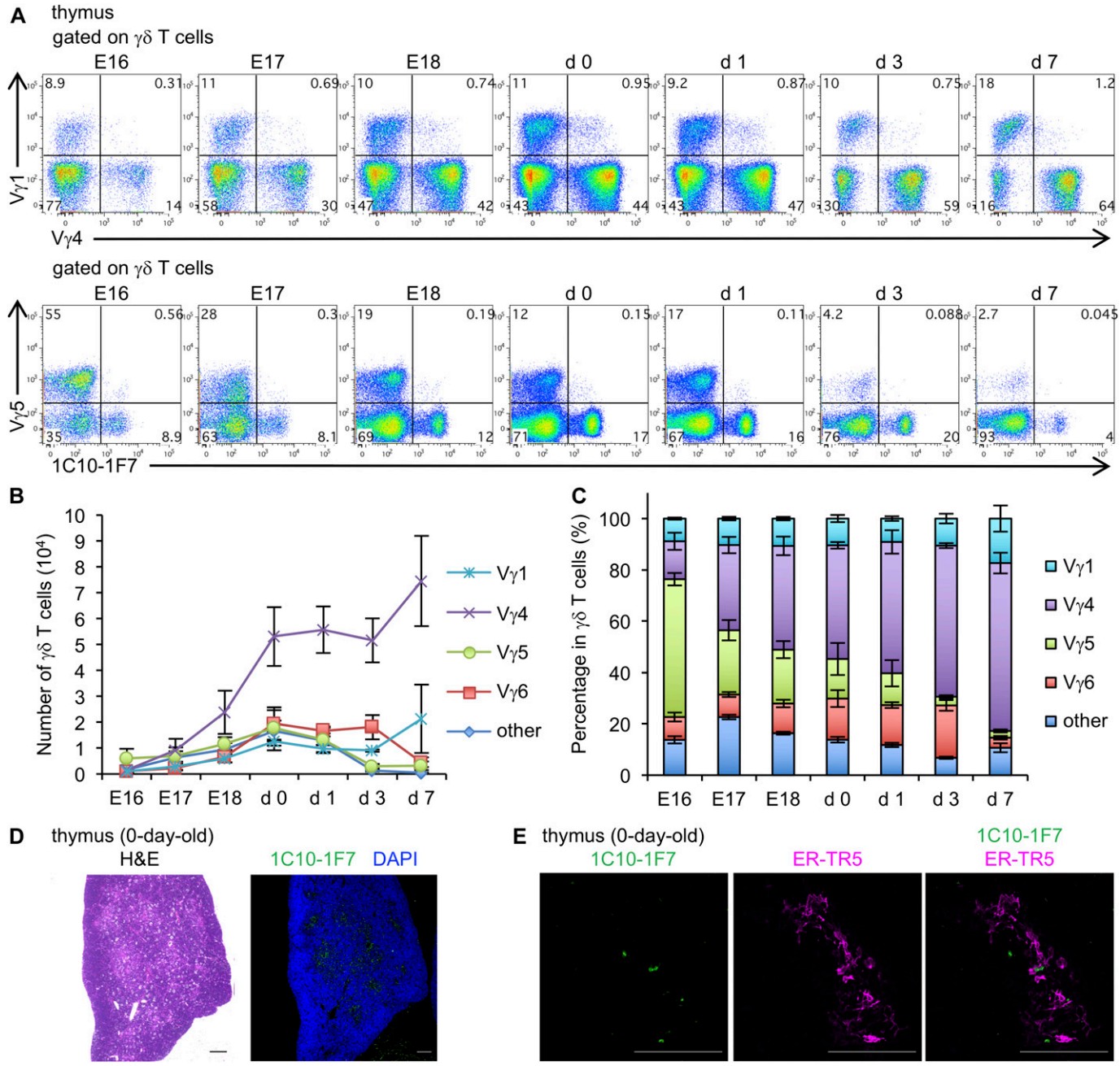

**Figure 3. Ontogenic wave of Vγ6 γδ T cells in thymus.**
**(A)** Representative dot plots are shown after gating γδ T cells. Numbers in quadrants indicate the percentage of expression of Vγ1 and Vγ4 (upper dot plots) or Vγ5 and 1C10-1F7 (lower dot plots) in the thymus from E16–7 d-old WT mice (n = 5). **(B)** Line graphs show the means ± SD of numbers of Vγ1⁺, Vγ4⁺, Vγ5⁺, Vγ6⁺, or other Vγ repertoire–positive γδ T cells in the thymus from E16–7-d-old WT mice (n = 5). **(C)** Bar graphs show the means ± SD of percentages of Vγ1⁺, Vγ4⁺, Vγ5⁺, Vγ6⁺, or other Vγ repertoire–positive γδ T cells in the thymus from E16–7-d-old WT mice (n = 5). **(D)** Paraformaldehyde-fixed paraffin section of thymus from 0-d-old WT mice was stained with H&E or Alexa Fluor 647–conjugated 1C10-1F7 (green) and DAPI (blue). **(E)** Acetone-fixed frozen section of thymus from 0-d-old WT mice was stained with Alexa Fluor 647–conjugated 1C10-1F7 (green) and ER-TR5 (magenta) mAbs. All scale bars represent 100 μm.

## Evaluation of in vivo effect of 1C10-1F7 mAb on Vγ6 γδ T cells

Paget et al (2015) reported that CD3$^{high}$ TCRδ$^{int}$ γδ T cells were IL-17A–producing Vγ6/Vδ1 γδ T cells and that CD3$^{int}$ TCRδ$^{high}$ γδ T cells were IL-17A–producing Vγ4 γδ T cells. We analyzed γδ T cells in the lungs from mice after they were inoculated intratracheally with

*Mycobacterium bovis* bacillus Calmette-Guérin (BCG) (Umemura et al, 2007) and stained with anti-CD3ε, TCRδ (clone: GL3), and 1C10-1F7 mAbs. Consistent with the previous report, we confirmed that IL-17A–producing CD3$^{high}$ TCRδ$^{int}$ γδ T cells were Vγ6⁺ γδ T cells and that IL-17A–producing CD3$^{int}$ TCRδ$^{high}$ γδ T cells were Vγ4⁺ γδ T cells (Fig 4A).

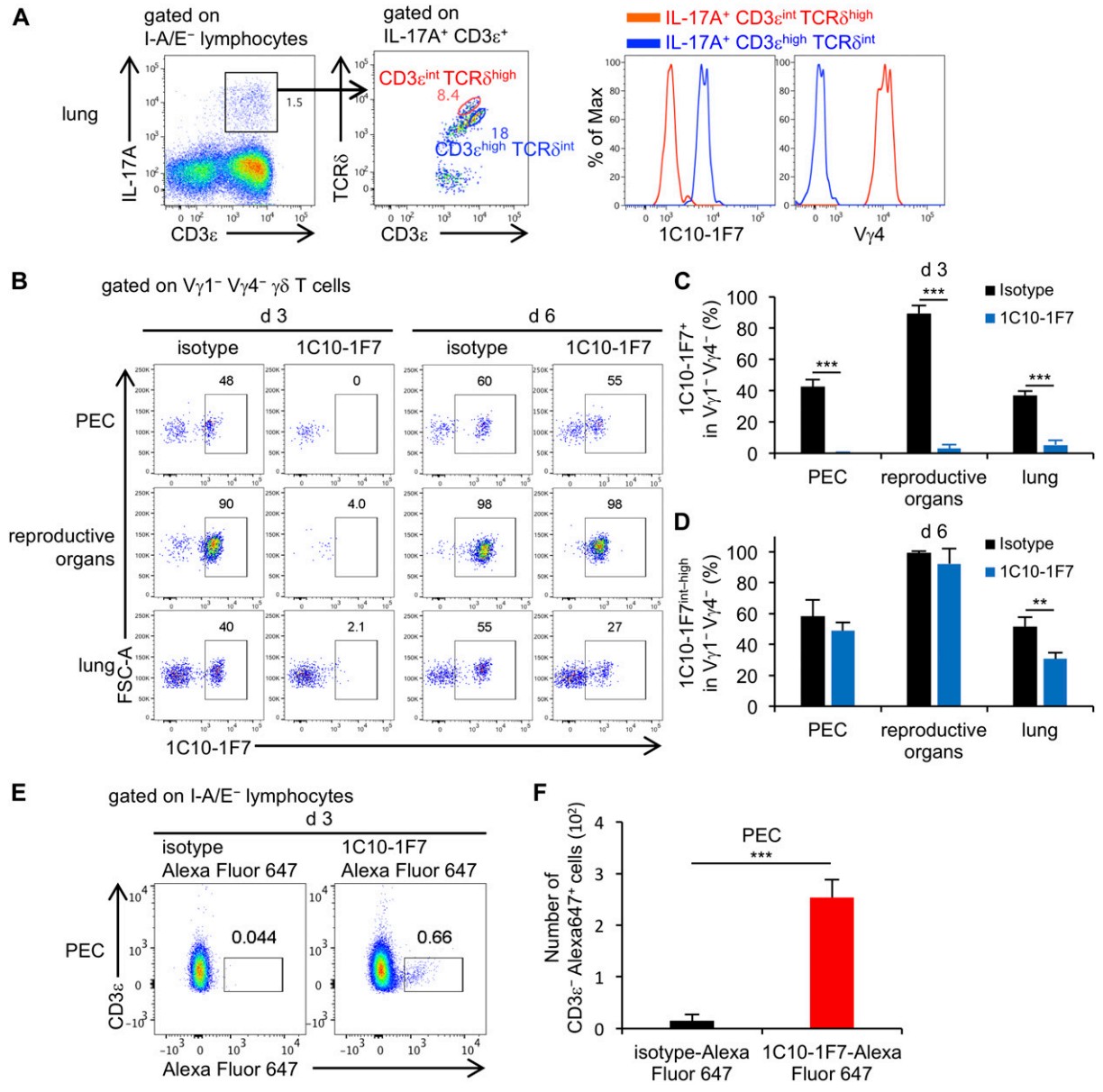

**Figure 4. Administration of 1C10-1F7 mAb in vivo may lead to internalized Vγ6 TCR.**
**(A)** Left dot plot shows after gating I-A/E⁻ lymphocytes and right dot plot shows post-gating of IL-17A⁺ CD3e⁺ cells in the lungs from BCG-infected mice after PAM/ionomycin stimulation. Histograms show expression of 1C10-1F7 staining and Vγ4 after gating IL-17A⁺ CD3ε^int TCRδ^high (red) or IL-17A⁺ CD3ε^high TCRδ^int (blue) cells. **(B)** Representative dot plots show post-gating Vγ1⁻ Vγ4⁻ γδ T cells from indicated organs on day 3 or 6 after intraperitoneal administration of 1C10-1F7 or mouse IgG1 isotype control mAbs. Numbers indicate the percentage of C10-1F7⁺ (d 3) or 1C10-1F7^int–high (d 6) (n = 3). **(C)** Bar graphs show the mean ± SD of the percentage of 1C10-1F7⁺ in Vγ1⁻ Vγ4⁻ γδ T cells of indicated organs on day 3 after administration of 1C10-1F7 or mouse IgG1 isotype control mAbs (n = 3). **(D)** Bar graphs show the mean ± SD of the percentage of 1C10-1F7^int–high in Vγ1⁻ Vγ4⁻ γδ T cells of indicated organs on day 6 after administration of 1C10-1F7 or mouse IgG1 isotype control mAbs (n = 3). **(E)** Representative dot plots show post-gating of I-A/E⁻ lymphocytes from PEC on day 3 after intraperitoneal administration of 1C10-1F7-Alexa Fluor 647 or mouse IgG1 isotype control-Alexa Fluor 647 mAbs. None of the antibodies used in the dot plots were conjugated to Alexa Fluor 647. **(F)** Bar graphs show the means ± SD of the numbers of CD3ε⁻ Alexa Fluor 647⁺ in I-A/E⁻ lymphocytes of PEC (n = 3). Significant differences are shown (*P < 0.05, **P < 0.01, and ***P < 0.001; using a t test).

We next examined the effect of in vivo administration of 1C10-1F7 mAb on Vγ6⁺ γδ T cells. Koenecke et al (2009) reported an in vivo application of mAb directed against γδ T cells (clone GL3, Armenian hamster IgG), leading to prolonged TCR internalization lasting at least 14 d, without clearance of the actual γδ T cells. As shown in Fig 4B–D, we found that Vγ6 TCR⁺ γδ T cells became invisible in the PEC, reproductive organs, and lungs on day 3 after in vivo administration

of 1C10-1F7 mAb (mouse IgG1, κ), whereas γδ T cells with a low intensity of Vγ6 TCR recovered in these organs by day 6 after administration (Fig 4B–D). To ensure that the 1C10-1F7 mAb is not a depleting mAb but is internalized by target cells, we used Alexa Fluor 647–conjugated 1C10-1F7 mAb for in vivo administration and found CD3⁻ Alexa Fluor 647⁺ cells, which internalized Vγ6 TCR, on day 3 after administration of Alexa Fluor 647–conjugated 1C10-1F7 mAb

(Fig 4E and F) (Koenecke et al, 2009). Because the IgG1 subclass does not bind to the mouse fc receptor IV and has no Ag-dependent cell-mediated cytotoxicity (ADCC) activity (Nimmerjahn & Ravetch, 2006), administration of 1C10-1F7 mAb (mouse IgG1, κ) in vivo might not delete but instead lead to the internalization of Vγ6 TCR on Vγ6 γδ T cells, lasting at least until day 6.

There exists extensive evidence of the involvement of IL-17A⁺ Vγ6⁺ γδ T cells in mounting an effective immune response against pathogens, including *Staphylococcal aureus* (Murphy et al, 2014), *Listeria monocytogenes* (Hamada et al, 2008; Sun et al, 2013), *Escherichia coli* (Shibata et al, 2007), *Bacillus subtilis* (Simonian et al, 2009), and *Mycobacterium tuberculosis* (Umemura et al, 2007; Guo et al, 2013). We recently reported that IL-17A⁺ Vγ1⁻ Vγ4⁻ γδ T cells expressing canonical Vγ6/Vδ1 genes were dominant over IL-17A⁺ Vγ4⁺ γδ T cells in the lungs of young mice after *Klebsiella pneumoniae* infection (Murakami et al, 2016). We then examined the in vivo effect of in vivo neutralization of Vγ6⁺ γδ T cells by administration of 1C10-1F7 mAb on the ability to protect against *K. pneumoniae* infection. We confirmed that Vγ6 γδ T cells in the lungs became invisible on day 3 after an intraperitoneal administration of 1C10-1F7 mAb (Fig 4B and C). As shown in Fig 5A and B, in vivo administration of 1C10-1F7 mAb resulted in impaired protection against *K. pneumoniae* infection (Fig 5A and B).

Further deleterious contributions of IL-17A–producing γδ T cells were observed in models of psoriasis (Chien et al, 2014), ischemic brain injury (Shichita et al, 2009), experimental autoimmune encephalomyelitis (Sutton et al, 2009), and collagen-induced arthritis (Roark et al, 2007). In all these cases, IL-17A⁺ γδ T cells are known to be major contributors of inflammation and associated disease pathology. Previous studies have demonstrated that IL-17A⁺ γδ T cells play a crucial role in psoriasis-like dermatitis induced by imiquimod (IMQ) (Cai et al, 2011, 2014; Gray et al, 2013). Upon IMQ treatment, IL-17A⁺ Vγ4⁺ γδ T cells specifically expand in the draining LN and recirculate to inflamed skin (Gray et al, 2013). However, we observed that IMQ-induced skin inflammation was significantly attenuated in mice that received 1C10-1F7 mAb (Fig 5C–G). These results indicate that at least Vγ6 γδ T cells contributed to pathogenesis of psoriasis such as dermatitis induced by IMQ. Constitutive TCRδ KO mice were reported to show similar IMQ pathology, whereas conditional TCRδ KO mice showed an attenuated pathology as compared with WT mice, suggesting that the pathological role of IL-17A⁺ γδ T cells may be compensated by other IL-17A⁺ cells in constitutive TCRδ KO mice (Sandrock et al, 2018). Block of Vγ6 TCR by in vivo administration of 1C10-1F7 mAb may be useful for investigation of the role of Vγ6 γδ T cells in various inflammatory diseases, similar to conditional TCRδ KO mice (Sandrock et al, 2018). However, the Vγ6 γδ T cells are still present after administration of

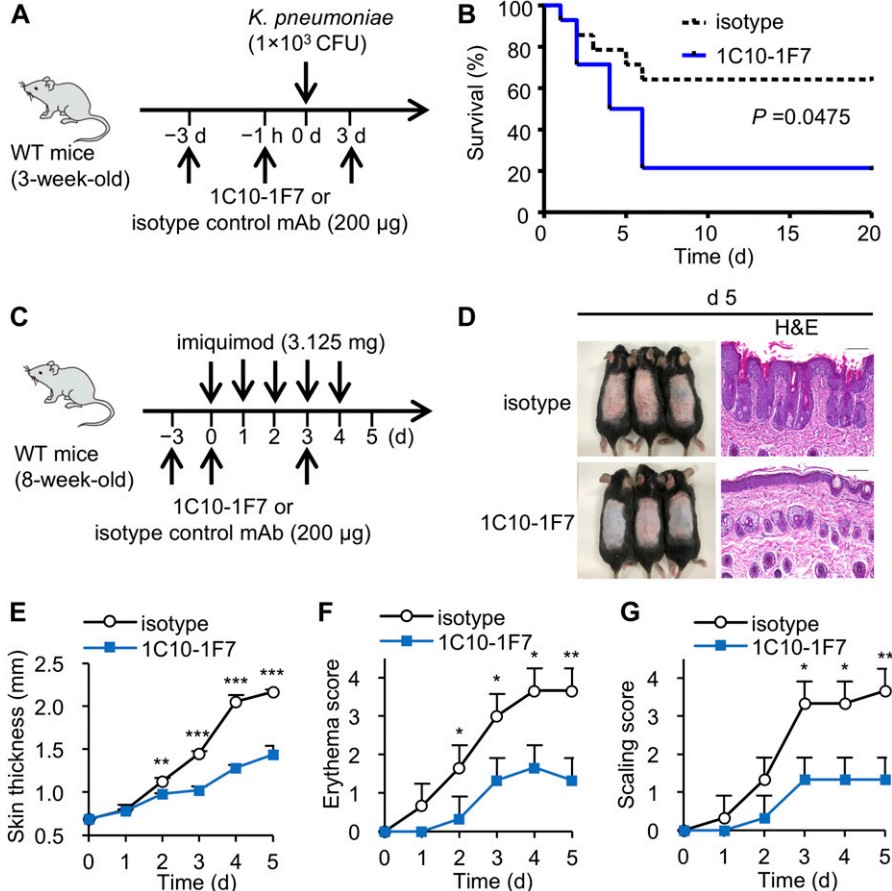

**Figure 5. 1C10-1F7 mAb is useful for elucidating the roles of Vγ6 γδ T cells in host defense.**
**(A, B)** 3-wk-old WT mice were intraperitoneally injected with 1C10-1F7 or mouse IgG1 isotype control mAbs on −3 d, −1 h, and 3 d. These mice were intranasally inoculated with *K. pneumoniae* at 1 × 10³ CFUs on day 0 (A), and survival was monitored every 24 h up to 20 days (n = 14 mice per group) (B). Data shown are combined from three different experiments. Statistical analyses of survival curves were performed by the log-rank test. Statistically significant differences are shown (*P* = 0.0475). **(C–G)** 8-wk-old WT mice were intraperitoneally injected with of 1C10-1F7 or isotype control mAbs on days −3, 0, and 3. These mice were applied daily 3.125 mg of IMQ on the shaved back on days 0–4. **(C)** Phenotypic presentation of mouse back skin (left panel) and H&E–stained sections of mouse back skin (right panel) were observed on day 5. Scale bars in H&E–stained sections represent 100 *μ*m. **(D)** Line graphs show the mean ± SD of back skin thickness on days 0–5. **(E)** Line graphs show the mean ± SD of erythema score (F) or scaling score (G) of the back skin on days 0–5 on a scale from 0 to 4 (n = 3). Significant differences are shown (*P* < 0.05, **P* < 0.01, and ***P* < 0.001 using a *t* test).

1C10-1F7 mAb and could potentially still react, for example, in a TCR-independent manner with cytokines via cytokine receptors and Toll-like receptors (Nakamura et al, 2008; Dejima et al, 2011).

In conclusion, we have successfully developed new mAbs specific to Vγ6⁺ γδ T cells. These mAbs are available for flow cytometry, immunohistochemistry, and in vivo function analysis. Vγ6⁺ γδ T cells play important roles in protection against microbial infection and in pathogenesis of inflammatory diseases such as colitis and autoimmune diseases by producing IL-17A (Chien et al, 2014). Our mAbs may be useful for elucidating the roles of Vγ6⁺ γδ T cells in these inflammatory diseases.

# Materials and Methods

### Mice

C57BL/6 or BALB/c female mice were purchased from Japan KBT. Vγ4/6 KO mice were generated as previously described (Sunaga et al, 1997). All mice were maintained under specific pathogen-free conditions and provided food and water ad libitum. Age- and gender-matched mice were used for all experiments. This study was approved by the Committee of Ethics on Animal Experiments of the Faculty of Medicine, Kyushu University. Experiments were carried out according to local guidelines for animal experimentation.

### Immunization and fusion protocols

For immunization, two types of peptides were synthesized from truncated regions of CDR1 ($V\gamma6_{21-35}$) and CDR2 ($V\gamma6_{50-66}$) in Vγ6 chains. Vγ4/6 KO mice were immunized with the KLH-conjugated peptides emulsified with CFA. These mice were reimmunized with the KLH-conjugated peptides without CFA 17 d after the first immunization. Iliac LNs were collected 21 d after the last immunization and were fused with SP2/0-Ag14 using polyethylene glycol.

### Cell lines

TG40 is a variant T-cell hybridoma cell line lacking the expression of TCR-$\alpha$ and TCR-$\beta$ chains, which has been used as recipient cells for TCR transfection (Sussman et al, 1988; Ohno et al, 1991). TG40 cell lines were introduced with the Vγ5Vδ1 or Vγ6Vδ1 genes using a retroviral bicistronic vector containing an internal ribosomal entry site (IRES) and rat CD2 (rCD2) (pMX–IRES–rCD2).

### Purification of new anti-Vγ6 mAbs

Hybridomas were cultured in Hybridoma-SFM (Thermo Fisher Scientific) including 1 ng/ml recombinant human IL-6 (R&D systems). After 7–10 d of culture, culture supernatants were collected. For flow cytometry and immunohistochemical analysis, new anti-Vγ6 mAbs were purified from the hybridoma supernatant using the mouse TCS purification system (Abcam) and conjugated with Alexa Fluor 647 using a labeling kit (Invitrogen). For the 1C10-1F7 administration experiment, 1C10-1F7 mAb was purified from the hybridoma supernatant using the Protein G Spin kit (Thermo Fisher Scientific).

### Sequence analysis of the heavy and light chain variable regions of Vγ6-specific mAbs

Total RNA was isolated and purified from hybridomas with RNeasy Plus Universal Mini Kit (QIAGEN). Total RNA was converted to complementary DNA (cDNA) with Superscript III reverse transcriptase (Invitrogen). Next, BCR genes were amplified using adaptor ligation-mediated PCR (Kitaura et al, 2017). High-throughput sequencing was performed using the Illumina Miseq paired-end platform (2 × 300 bp) (Illumina). V-D-J genes and CDR3 sequences were identified using IgBlast (NCBI: National Center for Biotechnology Information) (Ye et al, 2013).

### Cell preparations from various tissues

Single-cell suspensions were isolated from the thymus, PEC, reproductive organs (vagina/uterine cervix), i-IEL, c-LPL, and lungs as previously described (Shibata et al, 2008). Epidermal sheets were isolated from ears (Haas et al, 2012) and dETCs were isolated from the epidermal sheets by centrifugation at 600 $g$ for 20 min in a 40% and 70% Percoll (GE Healthcare Bio-Sciences AB) gradient.

### Flow cytometry analysis

Cells were stained for 20 min at 4°C with mAbs. We added 1 µg/ml propidium iodide (Sigma-Aldrich) to the cell suspension just before flow cytometry to detect and exclude dead cells from the surface staining analysis. To measure cytokine production, the cells were stimulated with 25 ng/ml PMA (Sigma-Aldrich) and 1 µg/ml ionomycin (Sigma-Aldrich) for 5 h at 37°C; 10 µg/ml Brefeldin A (Sigma-Aldrich) was added for the last 4 h of incubation. After the cells were stained with various mAbs, intracellular staining was performed according to the manufacturer's instructions (BD Biosciences). 100 µl BD Cytofix/Cytoperm solution (BD Biosciences) was added to the cell suspension with gentle mixing and incubated for 20 min at 4°C. Fixed cells were washed twice with 250 µl 10% BD Perm/Wash solution (BD Biosciences) and then stained intracellularly for 30 min at 4°C. Stained cells were analyzed on a FACSVerse flow cytometer (BD Biosciences) and data were analyzed using FlowJo software (Tree Star). Abs for flow cytometric analysis used in this study: PerCP-Cy5.5–conjugated anti-MHC class II (M5/114.15.2), APC-Cy7–conjugated anti-CD3ε (145-2C11), PE-Cy7–conjugated anti-CD45.2 (104), anti-Vγ4 (UC3-10A6) V500-conjugated anti-MHC class II (M5/114.15.2) mAbs, and streptavidin were purchased from BD Biosciences. PE-conjugated Hamster IgG isotype control (HTK888), PerCP-Cy5.5-conjugated anti-IL-17A (ebio17B7), biotin-conjugated anti-CD45.1 (A20), Purified Mouse IgG1 κ isotype control (P3), and Purified Mouse IgG2b κ isotype control (eBMG2b) mAbs were all purchased from eBioscience. FITC-conjugated, anti-MHC class II (M5/114.15.2), anti-Vγ1 (2.11), anti-Vγ4 (UC3-10A6), PE-conjugated anti-Vγ1 (2.11), anti-Vγ4 (UC3-10A6), anti-mouse IgG (Poly4053), APC-conjugated, anti-TCRδ (GL3), anti-Vγ4 (UC3-10A6), Alexa Fluor 647–conjugated anti-CD3ε (17A2), Mouse IgG1 κ isotype control

(MOPC-21), anti-mouse IgG (Poly4053), PE-Cy7–conjugated anti-TCRδ (GL3), V421-conjugated anti-TCRδ (GL3), and biotin-conjugated anti-rat IgM (MRM-47) mAbs were purchased from BioLegend. PE-conjugated anti-Vγ5 (536) mAb was purchased from Santa Cruz Biotechnology. Anti-Vγ7 (F2.67) and 17D1 mAbs were collected from F2.67 and 17D1 hybridoma culture supernatant.

### γδ T-cell sorting, RNA purification, RT-PCR, and sequencing of Vγ6

Single-cell suspensions were isolated from PEC and stained with mAbs. 1C10-1F7$^+$ γδ T cells were sorted using FACSAria (BD Bioscience). Total RNA was purified from sorted 1C10-1F7$^+$ γδ T cells using an RNeasy Mini kit (QIAGEN), and cDNA was synthesized using Superscript II (Invitrogen) according to the manufacturer's instructions. PCR was performed on a PCR thermal cycler (Takara Corp.). RT-PCR products were analyzed by blotting in 1.8% agarose gels. For RT-PCR analysis of Vγ6 TCR gene, combinations of following primers were used. Forward primers: Vγ6, 5′-GGAATTCAAAA-GAAAACATTGTCT-3′. Reverse primers: Cγ, 5′-CTTATGGAGATTTGTTTCAGC-3′. Forward primers: β-actin, 5′-TGGAATCCTGTGGCATCCATGAAAC-3′. Reverse primers: β-actin, 5′-TAAAACGCAGCTCAGTAACAGTCCG-3′. Purified Vγ6 PCR products of 1C10-1F7$^+$ γδ T cells from PEC were sequenced using BigDye Terminator v3.1 Cycle Sequencing kit (Applied Biosystems) and 3500xL Genetic Analyzers (Applied Biosystems).

### Immunohistochemistry

Uterine cervixes from 8-wk-old WT mice and thymus from 0-d-old WT mice were fixed with phosphate-buffered 4% paraformaldehyde (Nacalai Tesque) overnight and embedded in paraffin. Paraffin sections were stained with Alexa Fluor 647–conjugated 1C10-1F7 mAb, DAPI (BioLegend), and H&E. Frozen thymus from 0-d-old WT mice was embedded in OCT compound (Sakura Finetek) and frozen sections were fixed with acetone. Frozen sections were stained with Alexa Fluor 647–conjugated 1C10-1F7 mAb and ER-TR5 mAb followed by Alexa Fluor 488–conjugated anti-rat IgG (Thermo Fisher Scientific). For multicolor confocal analysis, slides were mounted in ProLong Gold Antifade reagent (Invitrogen) and analyzed with a Zeiss LSM700 confocal microscope (Carl Zeiss). H&E–stained slide was analyzed with All-in-One Fluorescence Microscope BZ-9000 (Keyence).

### Generation of BM and FL chimera

BM cells were extracted from 8-wk-old WT mice (Ly5.2/5.2) by flushing femurs and tibias and were then depleted of T cells using anti-CD3 mAb (17A2; BioLegend) and anti-rat IgG Dynabeads (Invitrogen). FL cells were extracted from the liver of embryonic day (ED) 14 WT mice (Ly5.2/5.2). $2 \times 10^7$ BM cells or $5 \times 10^6$ FL cells were intravenously injected into lethally irradiated (10 Gy) recipient 8-wk-old WT mice (Ly5.1/5.1). After 8 wk, reconstitution was confirmed.

### Microorganisms and bacterial infection experiment

Lyophilized *M. bovis* BCG (Tokyo strain) was purchased from Kyowa Pharmaceuticals and dissolved in 7H9 broth (Difco) supplemented with albumin–dextrose–catalase enrichment (BD Biosciences).

Single colonies were grown with vigorous shaking at 37°C in Middlebrook 7H9 broth supplemented with 10% albumin–dextrose–catalase, 1% glycerol (Sigma-Aldrich), and 0.5% Tween 80 (Wako) until the optical density at 600 nm ($OD_{600}$) reached 1. Bacteria were stored at –80°C in 50% glycerol as single-use aliquots. Mice were intratracheally infected with $1 \times 10^6$ CFUs of *M. bovis* BCG (Tokyo strain).

*K. pneumoniae* ATCC strain 43,816, serotype 2 (ATCC) was grown in Difco Nutrient Broth (Difco) for 18 h at 37°C with vigorous shaking. Bacteria were pelleted by centrifugation and stored at –80°C in 50% glycerol as single-use aliquots. 3-wk-old WT mice were intraperitoneally injected with 200 μg of 1C10-1F7 or mouse IgG1 isotype control mAbs (MOPC-21; Bioxcell) on –3 d, –1 h, and 3 d. These mice were intranasally inoculated with *K. pneumoniae* at $1 \times 10^3$ CFUs on day 0.

### IMQ experiments

8-wk-old WT mice were intraperitoneally injected with 200 μg of 1C10-1F7 or mouse IgG1 isotype control mAb on days –3, 0, and 3. These mice were applied daily a topical dose of 62.5 mg of commercially available IMQ cream (5%; Aldara; 3M Pharmaceuticals) on the shaved back for five consecutive days, translating to a daily dose of 3.125 mg of the active compound. Paraffin-fixed back skin of IMQ treatment on day 5 was fixed with phosphate-buffered 4% paraformaldehyde, embedded with paraffin, and stained with H&E. The back skin thickness was measured on days 0–5. The erythema score or scaling score of the back skin was scored on days 0–5 on a scale from 0 to 4.

### Statistical analysis

Statistical significance was evaluated using Prism software (GraphPad). The *t* test was used when only two groups were compared, and the survival curve was assessed by the log-rank test. *P* values <0.05 were considered to represent significant differences.

# Supplementary Information

# Acknowledgements

The authors are grateful to Dr. Y Matsumura and Dr. K Shibata for providing reagents and protocols and Y Kitada and A Yano for helping to prepare the manuscript. This work was supported by a Grant-in-Aid for Scientific Research on Innovative Areas JSPS KAKENHI no. JP 16H06496 and a Grant-in-Aid for JSPS Research Fellow no. JP 17J03389.

## Author Contributions

S Hatano: conceptualization, resources, data curation, software, formal analysis, supervision, funding acquisition, validation,

investigation, visualization, methodology, project administration, and writing—original draft, review, and editing.

X Tun: validation and investigation.

N Noguchi: validation and investigation.

D Yue: validation and investigation.

H Yamada: methodology.

X Sun: methodology.

M Matsumoto: resources and methodology.

Y Yoshikai: conceptualization, supervision, funding acquisition, methodology, project administration, and writing—original draft, review, and editing.

## Conflict of Interest Statement

The authors declare that they have no conflict of interest.

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
