## [Reviewer comments · Life Science Alliance]

Life Science Alliance

Development of a new monoclonal antibody specific to mouse Vy6 chain

Shinya Hatano, Xin Tun, Naoto Noguchi, Dan Yue, Hisakata Yamada, Xun Sun, Mitsuru Matsumoto, and Yasunobu Yoshikai

DOI: <https://doi.org/10.26508/lsa.201900363>

Corresponding author(s): Shinya Hatano, Division of Host Defense

Review Timeline:

Submission Date:	2019-03-01
Editorial Decision:	2019-04-05
Revision Received:	2019-04-24
Editorial Decision:	2019-04-26
Revision Received:	2019-04-28
Accepted:	2019-04-29

Scientific Editor: Andrea Leibfried

Transaction Report:

April 5, 2019

Re: Life Science Alliance manuscript #LSA-2019-00363-T

Dr. Shinya Hatano
Division of Host Defense
Medical Institute of Bioregulation
3-1-1 Maidashi, Higashi-ku
Fukuoka, Fukuoka 812-8582
Japan

Dear Dr. Hatano,

Thank you for submitting your manuscript entitled "Dichotomous roles of V γ 6 $\gamma\delta$ T cells in infection and inflammation in mice" to Life Science Alliance. The manuscript was assessed by expert reviewers, whose comments are appended to this letter.

As you will see, all three reviewers find your tool useful and of value to the community. They provide constructive input on how to slightly revise your manuscript to further strengthen it. We would thus like to invite you to submit a revised version for publication in Life Science Alliance.

Thank you for this interesting contribution to Life Science Alliance. We are looking forward to receiving your revised manuscript.

Sincerely,

Andrea Leibfried, PhD
Executive Editor
Life Science Alliance
Meyerhofstr. 1
69117 Heidelberg, Germany
t +49 6221 8891 502
e a.leibfried@life-science-alliance.org

B. MANUSCRIPT ORGANIZATION AND FORMATTING:

Reviewer #1 (Comments to the Authors (Required)):

This paper, "Dichotomous roles of Vg6 gamma delta T cells in infection and inflammation in mice," by Shinya Hatano, describes the generation of 3 monoclonal IgG antibodies that are specific for the murine TCR-Vg6 chain. This was accomplished by immunizing mice with a genetically ablated Vg6 gene (B6.Vg4/6-/- mice) with peptides representing specific regions of Vg6. All 3 antibodies recognize the CDR2 loop of this Vg, and the authors carefully show that detection of these cells using their new reagents agree with previous methods used for detecting cells expressing a Vg6+ TCR. The staining of Vg1+, Vg4+, Vg5+, and Vg6+ gd T cells during thymic development (Fig. 3) provides an especially nice addition to our understanding, as does the tissue localization studies showing a sub-epithelial location for Vg6+ cells in the female reproductive tract, and juxtaposed to mTECs in the neonatal thymic medulla during T cell development. The availability of these new mAbs will be likely be extremely useful in studies involving the Vg6+ gd T cells.

Addressing number of minor issues would improve the manuscript:

1. First, the English overall needs to be corrected - even the summary needs several corrections. In particular, the phrase in the summary "...played crucial roles in protection against *Klebsiella pneumoniae* infection but pathogenesis of psoriasis-like dermatitis," doesn't even make sense. Do the authors mean, "played crucial roles in protection against *Klebsiella pneumoniae* infection but were pathogenic in psoriasis-like dermatitis"? Perhaps they should add here "in agreement with earlier studies," since both of these were already reported.
2. The title of this paper should be changed, because roles for Vg6+ cells in infection and inflammation have already been shown in numerous studies. One suggestion for a new title might be, "Generation of monoclonal antibodies specific for TCR-invariant mouse Vg6+ cells"? I believe the immunofluorescence findings reported here are novel, and these could be instead or as well referred to in the title.
3. In the abstract, the authors mention TG40 but don't explain what this is. I suggest the authors mention their results with "transfectomas" instead.
4. On Line 34 - add which nomenclature is to be used in this paper (Heilig and Tonegawa).
5. On Line 44 - we now know of many more locations that the Vg6 Vd1 cells home to than are mentioned here - these should be added and referenced.
6. Line 55 seems a bit overstated - I don't believe it's really been shown that the Vg1/Vg2 specific mAb can actually stain Vg3+ cells.
7. On Line 56, please change the wording slightly for clarity: "which was first thought to detect dETCs bearing Vg5/Vd1 (Mallick-Wood et al., 1998), could bind Vg6/Vd1 gd T cells," change to, "which was first discovered to detect dETCs bearing Vg5/Vd1 (Mallick-Wood et al., 1998), could also bind Vg6/Vd1 gdT cells."
8. On Line 83, tiers should be titers.
9. On Line 88, please explain what TG40 is. It appears to be a TCR-deficient cell line. Are you using some sort of recombinant CD2 construct to transfect these cells? Explain the origin of this cell line, reference it, and briefly describe the constructs used (the construct description could be added in the methods section).
10. In Fig. 2E, the location of the Vg6+ cells actually appears to be sub-epithelial, just under the cervical epithelium.
11. Supplemental Fig. 2 needs to be fixed, it is too small to read and the background color makes this harder.
12. On Line 145, the author say: "Vg5+ gd T cells appeared first and decreased from E16 onwards during embryonic development;" but actually, they only start to decrease in number after d. 0 (Fig. 3B). Line 146 is also a little misleading, "The number of Vg6+ $\gamma\delta$ T cells increased gradually during embryonic development, reaching a peak at birth;" because the peak actually holds until after day 3 following birth. What is the gestation period in your facility? Please add this in the text - this could

make a difference - it can be longer in some locations and this could alter your results slightly from those previously published.

13. There is some discussion of the finding of Nitta et al. 2015 on line 165: "Nitta et al. recently reported that cortical thymic epithelial cells (cTEC) contribute to shaping the TCR Vg4 repertoire, and not Vg6 Tcells." This is not a very accurate synopsis of the findings of that paper, which rather showed that Vg4 IL17+ cells require mature cTECs to develop, whereas Vg6 IL17+ cells were substantially enhanced in mice that lack mature cTECs. In any case, these findings should be more thoroughly discussed in the discussion section instead. Are Vg5+ cells also found near the mTECs at a slightly earlier developmental stage?

14. In Fig. 4D, the authors show that prior injection of anti Vg6-specific mAb 1C10-1F7 conjugated to AF647 leads to the detection of AF647+ but CD3e-negative cells. This implies that the TCR has been internalized, but this needs to be explained better in the text.

15. On Line 189, the authors misquote a paper (Cho et al., 2010). This paper actually implicated Vg5+ cells, not Vg6+, in making an IL17 response against *Staphylococcus aureus*. There are other papers that could instead be quoted showing that *S. aureus* elicits a Vg6+ cell response, however (e.g. Murphy et al., *J. Immunol.* 2014).

16. On Line 235, please include references for the TG40 cell line and the retroviral vector used. It would be helpful to include a diagram of the retroviral TCR construct, for clarity (perhaps as a supplemental figure).

17. Reference Hatano et al. 2017 is inappropriate for the methods used for isolation of cells. Please replace this with appropriate references.

18. The methods section indicates that both C57BL/6 and BALB/c mice were used in these experiments. Please add to the figure legends which strains were used for the experiments shown.

Reviewer #2 (Comments to the Authors (Required)):

This study basically developed new monoclonal antibodies that specifically recognize mouse Vg6 T cells. Indeed in the gdT biology field, these mAbs are drastically needed although 17D1 mAb does provide a lot of helps in the past studies. The authors have presented convincing data showing that these mAbs recognize Vg6 T cells both in vitro and in vivo. Although no specific hypothesis was tested in this study, developing new reagents certainly will advance gdT cell biology study in the field. Overall the conclusions are well supported by the presented data. There are some minor comments for this manuscript.

1. There are some typos in the manuscript. For example, page 11 line 184, it should be "does" not "dose".

2. To ensure 1C10-1F7 mAb is not depleting mAb but is internalized by target cells, the authors may use fluorescent-labeled mAb to further test this.

Reviewer #3 (Comments to the Authors (Required)):

In this paper the authors describe the generation of three novel antibodies that are specific for the Vg6 chain expressed by the invariant Vg6Vd1 gd T cells. They used these antibodies for flow cytometry and immunohistochemistry in various organs. Importantly, they used one of these antibodies to investigate in vivo the role of these cells in models of bacterial infection and of psoriasis. Such a tool was lacking till now and thus can be very useful tool for the field in order to study various aspects of these cells, including their in vivo role and/or of the Vg6Vd1 TCR they express. However, as described in more detail below, the authors should discuss the usage of their antibody (antibodies?) compared to other possible tools to study Vg6+/gd T cells and some technical aspects need to be clarified,

1.

It seems that the 1C10-1F7 antibody (that the authors selected for their in vivo studies) rather internalise the Vg6+ gd TCR (and thus make the Vg6+ gd T cells 'invisible' for staining for Vg6) than depleting them. This is a major point that should be discussed, for example referring to the paper Koenecke 2009 EJI: 'In vivo application of mAb directed against the gammadelta TCR does not deplete but generates "invisible" gammadelta T cells', which is one of the references). So this antibody blocks rather the Vg6+ TCR in vivo? And thus the Vg6+ cells are still present and could potentially still react, for example via natural killer receptors and/or cytokine receptors?

In this context, the authors should also discuss the possible advantages (and disadvantages?) of using their Vg6 antibody vs other tools such as Vg6 -/- mice (replaced by other (gd) T cells?...). A recent paper in JEM (Sandrock et 2018) describing conditional depletion of gd T cells may be relevant as well.

In line 196 the authors state: 'We confirmed that Vg6 gd T cells in the lung were decreased on day 3 after an intraperitoneal administration of 1C10-1F7 mAb (Fig. 4, B and C).' But the authors suggest themselves that the Vg6 gd T cells are not decreased but the TCR is rather internalised. Please adapt.

2.

Line 92 (Introduction): 'The CDR loops on the membrane-distal face of TCRs comprise the site of ligand recognition. CDR3 loops are functionally critical for Ag recognition and CDR1 and CDR2 loops provide a perimeter of contacts surrounding a central region provided by CDR3 loops (Marrack et al., 2008). In the case of the V α 6 chain, the CDR2 α loop may be a perimeter surrounding the CDR1 α loop...'

This should be adapted. These observations are based on ab T cells. It is not sure at all that this is similar for gd T cells. It seems that gd T cells can recognise a wide range of (potential) ligands and different rules may exist for these different TCR-ligand combinations (reviewed recently in Vermijlen 2018 Sem in Cell and Dev Biology; see also Melandri et al 2018 NI)

3.

Why is the 1C10-1F7 antibody selected (out of three Vg6-specific antibodies generated) for their in vivo studies? This is not mentioned in the paper. Most efficient staining?

Also:

As a major point of the paper is the development of novel Vg6 antibodies, I suggest to reflect this in the title (for example: 'Dichotomous roles of Vg6 gd T cells in infection and inflammation in mice revealed by novel anti-Vg6 antibodies').

Line 180: '...while gd T cells with a low intensity of V α 6 TCR recovered in these organs by day 6 after

administration (Fig. 4, B-D).! This is not clear from me: where is this seen on the facs plots? Is the MFI of the 1C10-1F7-positive cells lower?

Line 22 in abstract: what is TG40? Is this important to mention in the abstract? Even in the Methods section (Line 236), it is not mentioned what 'TG40 cell lines' are. Please provide more details in the Methods section and reconsider the writing of the abstract.

Line 42 (Introduction): 'The second T cells to appear from E14 to birth carry V α 6 paired with V β 1 of $\alpha\alpha$ TCR (V α 6J α 1, V β 1D β 2J β 2) which homes to the epithelia of the reproductive tract (Itohara et al., 1990).! This should be updated (also lungs etc besides reproductive tract), as described further by the authors in their manuscript.

Line 49 (Introduction): This bias in V β usage has led to the suggestion that V β -encoded residues enable these T cells to respond to antigen (Ag) unique to their resident tissues.' Here the authors could refer to the recent paper in NI (Melandri et al 2018): The $\gamma\delta$ TCR combines innate immunity with adaptive immunity by utilizing spatially distinct regions for agonist selection and antigen responsiveness'.

Line 76: 'The most likely immunogenic epitopes for staining the Vg6 chain lie within the hypervariable CDR that provides the majority of binding contacts, so we selected peptides from CDR1 or CDR2 as immunogens'. Is the major reason not that the CDR3 of the Vg5 and Vg6 chain of the invariant Vg5Vd1 and Vg6Vd1 gd cells are the same?

Spelling:

Line 13: were located

Line 44: 'to' should be 'no'?

Line 83: titers

Reviewer #1 (Comments to the Authors (Required)):

This paper, "Dichotomous roles of Vg6 gamma delta T cells in infection and inflammation in mice," by Shinya Hatano, describes the generation of 3 monoclonal IgG antibodies that are specific for the murine TCR-Vg6 chain. This was accomplished by immunizing mice with a genetically ablated Vg6 gene (B6.Vg4/6-/- mice) with peptides representing specific regions of Vg6. All 3 antibodies recognize the CDR2 loop of this Vg, and the authors carefully show that detection of these cells using their new reagents agree with previous methods used for detecting cells expressing a Vg6+ TCR. The staining of Vg1+, Vg4+, Vg5+, and Vg6+ gd T cells during thymic development (Fig. 3) provides an especially nice addition to our understanding, as does the tissue localization studies showing a sub-epithelial location for Vg6+ cells in the female reproductive tract, and juxtaposed to mTECs in the neonatal thymic medulla during T cell development. The availability of these new mAbs will be likely be extremely useful in studies involving the Vg6+ gd T cells.

Addressing number of minor issues would improve the manuscript:

1. First, the English overall needs to be corrected - even the summary needs several corrections. In particular, the phrase in the summary "...played crucial roles in protection against *Klebsiella pneumoniae* infection but pathogenesis of psoriasis-like dermatitis," doesn't even make sense. Do the authors mean, "played crucial roles in protection against *Klebsiella pneumoniae* infection but were pathogenic in psoriasis-like dermatitis"? Perhaps they should add here "in agreement with earlier studies," since both of these were already reported.

Reply: Following the reviewer's suggestion, we changed the Summary blurb as follows:

"Using novel mAb specific to Vγ6TCR, we find Vγ6δγ⁺T cells are located in association with medullary thymic epithelial cells and play crucial roles in protection against *Klebsiella pneumoniae* infection but are pathogenic in psoriasis-like dermatitis in agreement with earlier studies." (Line 13–15, page 2 in revised manuscript)

2. The title of this paper should be changed, because roles for Vg6+ cells in infection and inflammation have already been shown in numerous studies. One suggestion for a new title might be, "Generation of monoclonal antibodies specific for TCR-invariant mouse Vg6+ cells"? I believe the immunofluorescence findings reported here are novel, and these could be instead or as well referred to in the title.

Reply: Following the reviewer's suggestion, we changed the Title as follows:

"Development of a new monoclonal antibody specific to mouse V γ 6 chain"

(Line 1, page 1 in revised manuscript)

3. In the abstract, the authors mention TG40 but don't explain what this is. I suggest the authors mention their results with "transfectomas" instead.

Reply: TG40 is a variant T cell hybridoma cell line lacking the expression of TCR- α and - β chains that has been used as recipient cells for TCR transfection (Ohno, H., C. Ushiyama, M. Taniguchi, R.N. Germain, and T. Saito. 1991. CD2 can mediate TCR/CD3-independent T cell activation. J. Immunol. 146:3742–374623).

Following the reviewer's suggestion, we changed the TG40 as follows in the abstract:

"T cell line without a cell-surface TCR"

(Line 22, page 2 in revised manuscript)

4. On Line 34 - add which nomenclature is to be used in this paper (Heilig and Tonegawa).

Reply: Following the reviewer's suggestion, we revised sentences as follows:

"The V γ genes are V γ 1, V γ 2, V γ 3, V γ 4, V γ 5, V γ 6, and V γ 7 using the Heilig & Tonegawa nomenclature (Heilig and Tonegawa, 1986) which we used here, or V γ 1.1, V γ 1.2, V γ 1.3, V γ 2, V γ 3, V γ 4, and V γ 5 using the Garman nomenclature (Garman et al., 1986)."

(Line 32–34, page 3 in revised manuscript)

5. On Line 44 - we now know of many more locations that the V γ 6 V δ 1 cells home to than are mentioned here - these should be added and referenced.

Reply: Following the reviewer's suggestion, we revised sentences and added a reference as follows:

“The second T cells to appear from E14 to birth carry V γ 6 paired with V δ 1 of $\gamma\delta$ TCR (V γ 6J γ 1, V δ 1D δ 2J δ 2) which homes to the epithelia of the reproductive tract, tongue, lung, peritoneal cavity (PEC), skin dermis, colon-lamina propria lymphocytes (c-LPL) and adipose tissue as tissue-associated cells (Itohara et al., 1990; Mokuno et al., 2000; Roark et al., 2004; Cai et al., 2011; Sun et al., 2013; Kohlgruber et al., 2018).”

(Line 42–46, page 3 in revised manuscript)

References

“Kohlgruber, A.C., S.T. Gal-Oz, N.M. LaMarche, M. Shimazaki, D. Duquette, H.F. Koay, H.N. Nguyen, A.I. Mina, T. Paras, A. Tavakkoli, U. von Andrian, A.P. Uldrich, D.I. Godfrey, A.S. Banks, T. Shay, M.B. Brenner, L. Lynch. 2018. $\gamma\delta$ T cells producing interleukin-17A regulate adipose regulatory T cell homeostasis and thermogenesis. *Nat Immunol* 19:464-474.”

(Line 466–469, page 28 in revised manuscript)

6. Line 55 seems a bit overstated - I don't believe it's really been shown that the V γ 1/V γ 2 specific mAb can actually stain V γ 3+ cells.

Reply: Following the reviewer's suggestion, we revised as follows:

“All mAbs specific to V γ chains, except for V γ 3 and V γ 6”

(Line 58, page 4 in revised manuscript)

7. On Line 56, please change the wording slightly for clarity: "which was first thought to detect dETCs bearing Vg5/Vd1 (Mallick-Wood et al., 1998), could bind Vg6/Vd1 gd T cells," change to, "which was first discovered to detect dETCs bearing Vg5/Vd1 (Mallick-Wood et al., 1998), could also bind Vg6/Vd1 gdT cells."

Reply: Following the reviewer's suggestion, we revised as follows:

"which was first thought to detect dETCs bearing V γ 5/V δ 1 (Mallick-Wood et al., 1998), could also bind V γ 6/V δ 1 $\gamma\delta$ T cells"

(Line 62–63, page 4 in revised manuscript)

8. On Line 83, tiers should be titers.

Reply: Following the reviewer's suggestion, we revised as follows:

"Ab titers"

(Line 88, page 6 in revised manuscript)

9. On Line 88, please explain what TG40 is. It appears to be a TCR-deficient cell line. Are you using some sort of recombinant CD2 construct to transfect these cells? Explain the origin of this cell line, reference it, and briefly describe the constructs used (the construct description could be added in the methods section).

Reply: Following the reviewer's suggestion, we revised sentences and added two references as follows:

"To further select mAbs available for cell surface staining, we screened for those mAbs capable of staining for TG40, is a cell-surface TCR-negative and intracytoplasmic CD3-positive mutant of the 21.2.2 mouse T cell line (Sussman et al., 1988; Ohno et al., 1991), which was introduced with V γ 6/V δ 1 genes (V γ 6V δ 1-rCD2) or with V γ 5/V δ 1 genes (V γ 5V δ 1-rCD2)."

(Line 92–95, page 6 in revised manuscript)

References

“Sussman, J.J., T. Saito, E. M. Shevach, R. N. Germain, J. D. Ashwell. 1988. Thy-1- and Ly-6-mediated lymphokine production and growth inhibition of a T cell hybridoma require co-expression of the T cell antigen receptor complex. *J Immunol* 140: 2520-2526.”

(Line 548–550, page 33 in revised manuscript)

“Ohno, H., C. Ushiyama, M. Taniguchi, R.N. Germain and T Saito. 1991. CD2 can mediate TCR/CD3-independent T cell activation. *J Immunol* 146:3742-6.”

(Line 503–504, page 30 in revised manuscript)

10. In Fig. 2E, the location of the $V\gamma 6^+$ cells actually appears to be sub-epithelial, just under the cervical epithelium.

Reply: We agreed with the reviewer’s suggestion. we revised as follows:

“the $V\gamma 6^+$ $\gamma\delta$ T cells were abundantly present at the sub-epithelial, just under the cervical epithelium (Fig. 2 E).”

(Line 141–142, page 9 in revised manuscript)

11. Supplemental Fig. 2 needs to be fixed, it is too small to read and the background color makes this harder.

Reply: Following the reviewer’s suggestion, we changed Supplemental Fig. 2 as follows:

Original Fig. S2

1C10-1F7⁺ $\gamma\delta$ T cells

	V γ Region	N Nucleotides	J γ Region
V $\gamma 6$ PCR products	CACWD	-	SSGFHKVF

After revised Fig.S2

1C10-1F7⁺ $\gamma\delta$ T cells

	V γ Region	N Nucleotides	J γ Region
V $\gamma 6$ PCR products	CACWD	-	SSGFHKVF

12. On Line 145, the author say: "V γ 5+ $\gamma\delta$ T cells appeared first and decreased from E16 onwards during embryonic development;" but actually, they only start to decrease in number after d. 0 (Fig. 3B). Line 146 is also a little misleading, "The number of V γ 6+ $\gamma\delta$ T cells increased gradually during embryonic development, reaching a peak at birth;" because the peak actually holds until after day 3 following birth. What is the gestation period in your facility? Please add this in the text - this could make a difference - it can be longer in some locations and this could alter your results slightly from those previously published.

Reply: We appreciate your comment. The gestation period is around 20th in our facility, like other facilities. Although the number of V γ 5+ $\gamma\delta$ T cells started to decrease after d 0, the percentage of V γ 5+ $\gamma\delta$ T cells in whole $\gamma\delta$ T cell population drastically decreased from E16. So, we revised sentences about V γ 5+ $\gamma\delta$ T cells as follows:

"We consistently found that V γ 5+ $\gamma\delta$ T cells were abundant at the earlier stage of fetal thymus and the percentage decreased from E16 onwards during embryonic development, while ensuing waves from E16 onward were V γ 4+ cells."

(Line 146–148, page 9 in revised manuscript)

We agree with reviewer's suggestion about V γ 6 $\gamma\delta$ T cells, we revised sentences about V γ 6 $\gamma\delta$ T cells as follow:

"The number of V γ 6+ $\gamma\delta$ T cells increased gradually during embryonic development, reaching a peak at neonatal stage from birth to day 3 (Fig. 3, A–C)."

(Line 148–150, page 9 in revised manuscript)

13. There is some discussion of the finding of Nitta et al. 2015 on line 165: "Nitta et al. recently reported that cortical thymic epithelial cells (cTEC) contribute to shaping the TCR V γ 4 repertoire, and not V γ 6 T cells." This is not a very accurate synopsis of the findings of that paper, which rather showed that V γ 4 IL17+ cells require mature cTECs to develop, whereas V γ 6 IL17+ cells were substantially enhanced in mice that lack mature cTECs. In any case, these findings should be more thoroughly discussed in the discussion section instead. Are V γ 5+ cells also found near the mTECs at a slightly earlier developmental stage?

Reply: Nitta et al. reported that in addition of lack of cTEC, mTEC are also reduced in TN mice deficient in the Psmb11 gene that encodes the cTEC-specific proteasome subunit b5t (Fig2c EMBO reports Vol 16 | No 5 | 2015). Therefore, we speculated that IL17+V γ 6 + T cells may be enriched due to reduced number of mTEC but not due to lack of cTEC in TN mice. Roberts et al. already described that in E17 fetal thymus, V γ 5+ thymocytes associate with developing Aire+ mTEC in Fig.1 (Immunity 36, 427–437, March 23, 2012). In this paper, signal form V γ 5+ thymocytes progenitor is essential for development of mTEC but mTEC may not shape V γ 5 repertoire. Therefore, we deleted the discussion on interaction of mTEC and V γ 5+T cells. According to the referee's suggestion, we discussed more about interaction of V γ 6⁺V δ 1⁺ T cells and mTEC as follows:

"It has been reported that in IL-17⁺ V γ 6⁺ V δ 1⁺ T cells are enriched in several organs of mice deficient in autoimmune regulator (Aire) gene which are expressed by mTEC (Fujikado et al., 2016). Nitta et al. recently reported that IL17⁺ V γ 6⁺ T cells were substantially enhanced in TN mice which have no mature cortical TEC (cTEC) and substantially reduced number of mTEC in thymus (Nitta et al., 2015). Taken together, it is suggested that mTEC negatively regulate the development of IL-17⁺ V γ 6⁺ γ δ T cells in thymus. However, this is only speculation and further experiments need to clarify the significance of interaction of V γ 6⁺ γ δ T cells and mTEC. On the other hand, Rank signaling has been reported to link the development of invariant V γ 5 γ δ T cell progenitors and the Aire⁺ medullary epithelium (Roberts et al., 2012). Thus, contribute to shaping the TCR V γ 5 and V γ 6 repertoire in a different manner."

(Line 164–171, page 10 in revised manuscript)

14. In Fig. 4D, the authors show that prior injection of anti V γ 6-specific mAb 1C10-1F7 conjugated to AF647 leads to the detection of AF647⁺ but CD3e-negative cells. This implies that the TCR has been internalized, but this needs to be explained better in the text.

Reply: Following the reviewer's suggestion, we added several sentences as follows:

“To ensure 1C10-1F7 mAb is not depleting mAb but is internalized by target cells, we used Alexa Fluor 647-conjugated 1C10-1F7 mAb for *in vivo* administration and found that CD3⁻ Alexa Fluor 647⁺ cells, which internalized V γ 6 TCR, on day 3 after administration of Alexa Fluor 647-conjugated 1C10-1F7 mAb (Fig. 4, E and F) (Koenecke et al., 2009).”

(Line 187–191, page 11–12 in revised manuscript)

15. On Line 189, the authors misquote a paper (Cho et al., 2010). This paper actually implicated V γ 5⁺ cells, not V γ 6⁺, in making an IL17 response against *Staphylococcus aureus*. There are other papers that could instead be quoted showing that *S. aureus* elicits a V γ 6⁺ cell response, however (e.g. Murphy et al., J. Immunol. 2014).

Reply: Following the reviewer's suggestion, we revised manuscript and references as follows:

“There exists extensive evidence of the involvement of IL-17A⁺ V γ 6⁺ $\gamma\delta$ T cells in mounting an effective immune response against pathogens, including *Staphylococcal aureus* (Cho et al., 2010 Murphy et al., 2014)”

(Line 195–196, page 12 in revised manuscript)

References

~~Cho, J.S., E.M. Pietras, N.C. Garcia, R.I. Ramos, D.M. Farzam, H.R. Monroe, J.E. Magorien, A. Blauvelt, J.K. Kolls, A.L. Cheung, G. Cheng, R.L. Modlin, and L.S. Miller. 2010. IL-17 is essential for host defense against cutaneous *Staphylococcus aureus* infection in mice. *J Clin Invest* 120:1762–1773.~~

Murphy, A.G., K.M. O’Keeffe, S.J. Lalor, B.M. Maher, K.H. Mills and R.M. McLoughlin. 2014. *Staphylococcus aureus* Infection of Mice Expands a Population of Memory $\gamma\delta$ T Cells

That Are Protective against Subsequent Infection. *J Immunol* 192:3697-708.”

(Line 491–493, page 29 in revised manuscript)

16. On Line 235, please include references for the TG40 cell line and the retroviral vector used. It would be helpful to include a diagram of the retroviral TCR construct, for clarity (perhaps as a supplemental figure).

Reply: We revised sentences and added two references as follows:

“TG40 is a variant T cell hybridoma cell line lacking the expression of TCR- α and - β chains that has been used as recipient cells for TCR transfection (Sussman et al., 1988; Ohono et al., 1991). TG40 cell lines were introduced with the V γ 5V δ 1 or V γ 6V δ 1 genes using a retroviral bicistronic vector containing an internal ribosomal entry site (IRES) and rat CD2 (rCD2) (pMX–IRES–rCD2).”

(Line 249–252, page 16 in revised manuscript)

References

“Sussman, J.J., T. Saito, E. M. Shevach, R. N. Germain, J. D. Ashwell. 1988. Thy-1- and Ly-6-mediated lymphokine production and growth inhibition of a T cell hybridoma require co-expression of the T cell antigen receptor complex. *J Immunol* 140: 2520-2526.”

(Line 548–550, page 32 in revised manuscript)

“Ohno, H., C. Ushiyama, M. Taniguchi, R.N. Germain and T Saito. 1991. CD2 can mediate TCR/CD3-independent T cell activation. *J Immunol* 146:3742-6.”

(Line 503–504, page 30 in revised manuscript)

17. Reference Hatano et al. 2017 is inappropriate for the methods used for isolation of cells. Please replace this with appropriate references.

Reply: Following the reviewer's suggestion, we revised methods for isolation of cells and references as follows:

“Single-cell suspensions were isolated from the thymus, PEC, reproductive organs (vagina/uterine cervix), i-IEL, c-LPL and lung as previously described (Shibata et al., 2008). Epidermal sheets were isolated from ears (Haas et al., 2012) and dETCs were isolated from the epidermal sheets by centrifugation at 600 x g for 20 min in a 40% and 70% Percoll (GE Healthcare Bio-Sciences AB, Uppsala, Sweden) gradient.”

(Line 272–276, page 17 in revised manuscript)

“References

Hatano, S., T. Murakami, N. Noguchi, H. Yamada, and Y. Yoshikai. 2017. CD5⁻NK1.1⁺ $\gamma\delta$ T cells that develop in a *Bel11b*-independent manner participate in early protection against infection. *Cell Rep* 21:1191-1202.

“Haas, J.D., S. Ravens, S. Düber, I. Sandrock, L. Oberdörfer, E. Kashani, V. Chennupati, L. Föhse, R. Naumann, S. Weiss, A. Krueger, R. Förster and I. Prinz. 2012. Development of interleukin-17-producing $\gamma\delta$ T cells is restricted to a functional embryonic wave. *Immunity* 37:48-59.”

(Line 431–433, page 26 in revised manuscript)

“Shibata, K., H. Yamada, R. Nakamura, X. Sun, M. Itsumi and Y. Yoshikai. 2008. Identification of CD25⁺ gamma delta T cells as fetal thymus-derived naturally occurring IL-17 producers. *J Immunol* 181:5940–5947.”

(Line 529–531, page 31–32 in revised manuscript)

18. The methods section indicates that both C57BL/6 and BALB/c mice were used in these experiments. Please add to the figure legends which strains were used for the experiments shown.

Reply: Following the reviewer's suggestion, we added mice strain in figure legends.

Reviewer #2 (Comments to the Authors (Required)):

This study basically developed new monoclonal antibodies that specifically recognize mouse V γ 6 T cells. Indeed in the $\gamma\delta$ T biology field, these mAbs are drastically needed although 17D1 mAb does provide a lot of helps in the past studies. The authors have presented convincing data showing that these mAbs recognize V γ 6 T cells both in vitro and in vivo. Although no specific hypothesis was tested in this study, developing new reagents certainly will advance $\gamma\delta$ T cell biology study in the field. Overall the conclusions are well supported by the presented data. There are some minor comments for this manuscript.

1. There are some typos in the manuscript. For example, page 11 line 184, it should be "does" not "dose".

Reply: Following the reviewer's suggestion, we revised typos throughout the text:

“Since the IgG1 subclass does not bind” (Line 191, page 12 in revised manuscript)

“even no nucleotides in the TCR gene junction and are essentially an oligoclonal population of cells.” (Line 47, page 4 in revised manuscript)

“Ab titers” (Line 88, page 6 in revised manuscript)

2. To ensure 1C10-1F7 mAb is not depleting mAb but is internalized by target cells, the authors may use fluorescent-labeled mAb to further test this.

Reply: We used Alexa Fluor 647-conjugated 1C10-1F7 mAb in Fig. 4, E and F.

We further added the following sentences to

“To ensure 1C10-1F7 mAb is not depleting mAb but is internalized by target cells, we used Alexa Fluor 647-conjugated 1C10-1F7 mAb for *in vivo* administration and found that CD3⁻ Alexa Fluor 647⁺ cells, which internalized V γ 6 TCR, on day 3 after administration of Alexa Fluor 647-conjugated 1C10-1F7 mAb (Fig. 4, E and F) (Koenecke et al., 2009).”

(Line 187–191, page 11–12 in revised manuscript)

Reviewer #3 (Comments to the Authors (Required)):

In this paper the authors describe the generation of three novel antibodies that are specific for the V γ 6 chain expressed by the invariant V γ 6V δ 1 $\gamma\delta$ T cells. They used these antibodies for flow cytometry and immunohistochemistry in various organs. Importantly, they used one of these antibodies to investigate in vivo the role of these cells in models of bacterial infection and of psoriasis. Such a tool was lacking till now and thus can be very useful tool for the field in order to study various aspects of these cells, including their in vivo role and/or of the V γ 6V δ 1 TCR they express. However, as described in more detail below, the authors should discuss the usage of their antibody (antibodies?) compared to other possible tools to study V γ 6+/ $\gamma\delta$ T cells and some technical aspects need to be clarified,

1. It seems that the 1C10-1F7 antibody (that the authors selected for their in vivo studies) rather internalise the V γ 6+ gd TCR (and thus make the V γ 6+ gd T cells 'invisible' for staining for V γ 6) than depleting them. This is a major point that should be discussed, for example referring to the paper Koenecke 2009 EJI: 'In vivo application of mAb directed against the gammadelta TCR does not deplete but generates "invisible" gammadelta T cells', which is one of the references). So this antibody blocks rather the V γ 6+ TCR in vivo? And thus the V γ 6+ cells are still present and could potentially still react, for example via natural killer receptors and/or cytokine receptors?

In this context, the authors should also discuss the possible advantages (and disadvantages?) of using their V γ 6 antibody vs other tools such as V γ 6 -/- mice (replaced by other (gd) T cells?...). A recent paper in JEM (Sandrock et 2018) describing conditional depletion of gd T cells may be relevant as well. In line 196 the authors state: 'We confirmed that V γ 6 gd T cells in the lung were decreased on day 3 after an intraperitoneal administration of 1C10-1F7 mAb (Fig. 4, B and C).' But the authors suggest themselves that the V γ 6 gd T cells are not decreased but the TCR is rather internalised. Please adapt.

#1-1 It seems that the 1C10-1F7 antibody (that the authors selected for their *in vivo* studies) rather internalise the V γ 6+ $\gamma\delta$ TCR (and thus make the V γ 6+ $\gamma\delta$ T cells 'invisible' for staining for V γ 6) than depleting them. This is a major point that should be discussed, for example referring to the paper Koenecke 2009 EJI: 'In vivo application of mAb directed against the gammadelta TCR does not deplete but generates "invisible" gammadelta T cells', which is one of the references). So this antibody blocks rather the V γ 6+ TCR *in vivo*? And thus the V γ 6+ cells are still present and could potentially still react, for example via natural killer receptors and/or cytokine receptors?

Reply: We think that 1C10-1F7 antibody rather internalize the V γ 6⁺ $\gamma\delta$ TCR than depleting them and thus make the V γ 6+ $\gamma\delta$ T cells 'invisible' for staining for V γ 6. We discussed this points and revised sentences as follows:

“We next examined the effect of *in vivo* administration of 1C10-1F7 mAb on V γ 6⁺ $\gamma\delta$ T cells. Koenecke *et al.* reported an *in vivo* application of mAb directed against $\gamma\delta$ T cells (clone GL3, Armenian hamster IgG), leading to prolonged TCR internalization lasting at least 14 days, without clearance of the actual $\gamma\delta$ T cells (Koenecke *et al.*, 2009). As shown in Figures 4B to 4D, we found that V γ 6 TCR⁺ $\gamma\delta$ T cells became invisible in the PEC, reproductive organs and lung on day 3 after *in vivo* administration of 1C10-1F7 mAb (mouse IgG1, κ), while $\gamma\delta$ T cells with a low intensity of V γ 6 TCR recovered in these organs by day 6 after administration (Fig. 4, B–D). To ensure 1C10-1F7 mAb is not depleting mAb but is internalized by target cells, we used Alexa Fluor 647-conjugated 1C10-1F7 mAb for *in vivo* administration and found that CD3⁻ Alexa Fluor 647⁺ cells, which internalized V γ 6 TCR, on day 3 after administration of Alexa Fluor 647-conjugated 1C10-1F7 mAb (Fig. 4, E and F) (Koenecke *et al.*, 2009).”

(Line 181–191, page 11–12 in revised manuscript)

#1-2 In this context, the authors should also discuss the possible advantages (and disadvantages?) of using their V γ 6 antibody vs other tools such as V γ 6^{-/-} mice (replaced by other ($\gamma\delta$) T cells?...). A recent paper in JEM (Sandrock *et al.* 2018) describing conditional depletion of gd T cells may be relevant as well.

Reply: According the referee's comment, we discussed this point and revised sentences as follows:

“Constitutive TCR δ KO mice were reported to show similar IMQ pathology, while conditional TCR δ KO mice showed an attenuated pathology as compared with WT mice, suggesting that the pathological role of IL17A⁺ $\gamma\delta$ T cells may be compensated by other IL17A⁺ cells in constitutive TCR δ KO mice (Sandrock et al., 2018). Block of V γ 6 TCR by *in vivo* administration of 1C10-1F7 mAb may be useful for investigation of the role of V γ 6 $\gamma\delta$ T cells in various inflammatory diseases, similar to conditional TCR δ KO mice (Sandrock et al., 2018). However, the V γ 6 $\gamma\delta$ T cells are still present after administration of 1C10-1F7 mAb and could potentially still react, for example in TCR independent manner with cytokines via cytokine receptors and Toll-like receptors (Nakamura et al., 2008; Dejima et al., 2011)
(Line 217–225, page 13 in revised manuscript)

We added a reference as follows:

References

“Sandrock, I., A. Reinhardt, S. Ravens, C. Binz, A. Wilharm, J. Martins, L. Oberdörfer, L. Tan, S. Lienenklaus, B. Zhang, R. Naumann, Y. Zhuang, A. Krueger, R. Förster and I. Prinz. 2018. Genetic models reveal origin, persistence and non-redundant functions of IL-17–producing $\gamma\delta$ T cells. *J Exp Med* 215:3006-3018.”

(Line 522–525, page 31 in revised manuscript)

“Nakamura, R., K. Shibata, H. Yamada, K. Shimoda, K. Nakayama and Y. Yoshikai 2008. Tyk2-signaling plays an important role in host defense against *Escherichia coli* through IL-23-induced IL-17 production by $\gamma\delta$ T cells. *J Immunol* 181:2071-5.”

(Line 494–496, page 29 in revised manuscript)

“Dejima, T., K. Shibata, H. Yamada, H. Hara, Y. Iwakura, S. Naito and Y. Yoshikai. 2011. A protective role of naturally occurring IL-17A-producing $\gamma\delta$ T cells in the lung at the early stage of systemic candidiasis in mice. *Infect Immun* 79:4503-10.”

(Line 404–406, page 25 in revised manuscript)

#1-3 In line 196 the authors state: 'We confirmed that V γ 6 $\gamma\delta$ T cells in the lung were decreased on day 3 after an intraperitoneal administration of 1C10-1F7 mAb (Fig. 4, B and C).' But the authors suggest themselves that the V γ 6 $\gamma\delta$ T cells are not decreased but the TCR is rather internalised. Please adapt.

Reply: We changed the sentences as follows:

“We confirmed that V γ 6 $\gamma\delta$ T cells in the lung became invisible on day 3 after an intraperitoneal administration of 1C10-1F7 mAb (Fig. 4, B and C).”

(Line 203–204, page 12 in revised manuscript)

2.

Line 92 (Introduction): 'The CDR loops on the membrane-distal face of TCRs comprise the site of ligand recognition. CDR3 loops are functionally critical for Ag recognition and CDR1 and CDR2 loops provide a perimeter of contacts surrounding a central region provided by CDR3 loops (Marrack et al., 2008). In the case of the V γ 6 chain, the CDR2 γ loop may be a perimeter surrounding the CDR1 γ loop....'

This should be adapted. These observations are based on ab T cells. It is not sure at all that this is similar for gd T cells. It seems that $\gamma\delta$ T cells can recognize a wide range of (potential) ligands and different rules may exist for these different TCR-ligand combinations (reviewed recently in Vermijlen 2018 Sem in Cell and Dev Biology; see also Melandri et al 2018 NI)

Reply:

We agree with the reviewer's comment. It is only speculation from the data based on TCR $\alpha\beta$ structure. In introduction, we deleted as follows; “~~The CDR loops on the membrane-distal face of TCRs comprise the site of ligand recognition. CDR3 loops are functionally critical for Ag recognition and CDR1 and CDR2 loops provide a perimeter of contacts surrounding a central region provided by CDR3 loops (Marrack et al., 2008). In the case of the V γ 6 chain, the CDR2 γ loop may be a perimeter surrounding the CDR1 γ loop.~~”

We deleted a reference as follows;

“References

~~Marraek, P., J.P. Scott-Browne, S. Dai, L. Gapin, and J.W. Kappler. 2008. Evolutionarily conserved amino acids that control TCR-MHC interaction. *Annu Rev Immunol* 26:171-203.”~~

3. Why is the 1C10-1F7 antibody selected (out of three V γ 6-specific antibodies generated) for their in vivo studies? This is not mentioned in the paper. Most efficient staining?

Reply: We selected 1C10-1F7 mAb (IgG1, κ) with the highest affinity in further experiments.
We added this sentence in line 127–128, page 8 in revised manuscript.
(Line 127–128, page 8 in revised manuscript)

Also: As a major point of the paper is the development of novel V γ 6 antibodies, I suggest to reflect this in the title (for example: 'Dichotomous roles of V γ 6 $\gamma\delta$ T cells in infection and inflammation in mice revealed by novel anti-V γ 6 antibodies').

Reply: Following the reviewer’s suggestion, we changed the Title as follows:

“Development of a new monoclonal antibody specific to mouse V γ 6 chain”
(Line 1, page 1 in revised manuscript)

Line 180: '...while $\gamma\delta$ T cells with a low intensity of V γ 6 TCR recovered in these organs by day 6 after administration (Fig. 4, B-D).' This is not clear from me: where is this seen on the facs plots? Is the MFI of the 1C10-1F7-positive cells lower?

Reply: Please see FACS plots on d6 after administration 1C10-1F7 mAb and isotype IgG in Fig 4B. The MFI of 1C10-1F7⁺ cells on d6 after administration 1C10-1F7 mAb were lower than the MFI of 1C10-1F7⁺ cells on d6 after administration isotype IgG.

Line 22 in abstract: what is TG40? Is this important to mention in the abstract? Even in the Methods section (Line 236), it is not mentioned what 'TG40 cell lines' are. Please provide more details in the Methods section and reconsider the writing of the abstract.

Reply: Following the reviewer's suggestion, we changed the TG40 as follows in the abstract:

“T cell line without a cell-surface TCR”

(Line 22, page 2 in revised manuscript)

We revised sentences and added two references as follows:

“To further select mAbs available for cell surface staining, we screened for those mAbs capable of staining for TG40, is a cell-surface TCR-negative and intracytoplasmic CD3-positive mutant of the 21.2.2 mouse T cell line (Sussman et al., 1988; Ohno et al., 1991), which was introduced with V γ 6/V δ 1 genes (V γ 6V δ 1-rCD2) or with V γ 5/V δ 1 genes (V γ 5V δ 1-rCD2).”

(Line 92–95, page 6 in revised manuscript)

References

“Sussman, J.J., T. Saito, E. M. Shevach, R. N. Germain, J. D. Ashwell. 1988. Thy-1- and Ly-6-mediated lymphokine production and growth inhibition of a T cell hybridoma require co-expression of the T cell antigen receptor complex. *J Immunol* 140: 2520-2526.”

(Line 548–550, page 32 in revised manuscript)

“Ohno, H., C. Ushiyama, M. Taniguchi, R.N. Germain and T Saito. 1991. CD2 can mediate TCR/CD3-independent T cell activation. *J Immunol* 146:3742-6.”

(Line 503–504, page 30 in revised manuscript)

We revised materials and methods in detail as follows;

“TG40 is a variant T cell hybridoma cell line lacking the expression of TCR- α and - β chains that has been used as recipient cells for TCR transfection (Sussman et al., 1988; Ohono et al., 1991). TG40 cell lines were introduced with the V γ 5V δ 1 or V γ 6V δ 1 genes using a retroviral bicistronic vector containing an internal ribosomal entry site (IRES) and rat CD2 (rCD2) (pMX-IRES-rCD2).”

(Line 249-252, page 16 in revised manuscript)

Line 42 (Introduction): 'The second T cells to appear from E14 to birth carry V γ 6 paired with V δ 1 of $\gamma\delta$ TCR (V γ 6J γ 1, V δ 1D δ 2J δ 2) which homes to the epithelia of the reproductive tract (Itohara et al., 1990).' This should be updated (also lungs etc besides reproductive tract), as described further by the authors in their manuscript.

Reply: Following the reviewer’s suggestion, we revised sentences and added a reference as follows:

“The second T cells to appear from E14 to birth carry V γ 6 paired with V δ 1 of $\gamma\delta$ TCR (V γ 6J γ 1, V δ 1D δ 2J δ 2) which homes to the epithelia of the reproductive tract, tongue, lung, peritoneal cavity (PEC), skin dermis, colon-lamina propria lymphocytes (c-LPL) and adipose tissue as tissue-associated cells (Itohara et al., 1990; Mokuno et al., 2000; Roark et al., 2004; Cai et al., 2011; Sun et al., 2013; Kohlgruber et al., 2018).”

(Line 42–46, page 3 in revised manuscript)

References

“Kohlgruber, A.C., S.T. Gal-Oz, N.M. LaMarche, M. Shimazaki, D. Duquette, H.F. Koay, H.N. Nguyen, A.I. Mina, T. Paras, A. Tavakkoli, U. von Andrian, A.P. Uldrich, D.I. Godfrey, A.S. Banks, T. Shay, M.B. Brenner, L. Lynch. 2018. $\gamma\delta$ T cells producing interleukin-17A regulate adipose regulatory T cell homeostasis and thermogenesis. *Nat Immunol* 19:464-474.”

(Line 466–470, page 28 in revised manuscript)

Line 49 (Introduction): This bias in V γ usage has led to the suggestion that V γ -encoded residues enable these T cells to respond to antigen (Ag) unique to their resident tissues.' Here the authors could refer to the recent paper in NI (Melandri et al 2018): The $\gamma\delta$ TCR combines innate immunity with adaptive immunity by utilizing spatially distinct regions for agonist selection and antigen responsiveness'.

Reply: We appreciate your suggestion. We added the following sentences in Introduction section and cited the reference in the reference section.

Recently, V γ 7⁺ i-IEL are reported to response to epithelial butyrophilin -like (Btl) protein of the B7 superfamily using germline-encoded motifs distinct from CDRs within the V γ 7 chain (Melandri et al., 2018; Di Marco Barros et al., 2016). Thus, the bias of V γ usage in various mucosal tissues has led to the suggestion that V γ -encoded residues enable these T cells to respond to agonists unique to their resident tissues.

(Line 53–57, page 4 in revised manuscript)

Reference

“Melandri, D., I. Zlatareva, R.A.G. Chaleil, R.J. Dart, A. Chancellor, O. Nussbaumer, O. Polyakova, N.A. Roberts, D. Wesch, D. Kabelitz, P.M. Irving, S. John, S. Mansour, P.A. Bates, P Vantourout and A. C. Hayday. 2018. The $\gamma\delta$ TCR combines innate immunity with adaptive immunity by utilizing spatially distinct regions for agonist selection and antigen responsiveness. *Nat Immunol* 19:1352-1365.”

(Line 476–480, page 29 in revised manuscript)

“Di Marco Barros, R., N.A. Roberts, R.J. Dart, P. vantourout, A. Jamdke, O. Nussbaumer, L.Deban, S. Cipolat, R. Hart, M.L. Lannitto, A. Laing, B. Spencer-Dene, D. Gibboms, P.M. Lrving, P Pereira, U. Steinhoff, A. Hayday. 2016. Epithelia Use Butyrophilin-like Molecules to Shape Organ-Specific $\gamma\delta$ T Cell Compartments. *Cell* 167:203-218.”

(Line 407–410, page 25 in revised manuscript)

Line 76: 'The most likely immunogenic epitopes for staining the V γ 6 chain lie within the hypervariable CDR that provides the majority of binding contacts, so we selected peptides from

CDR1 or CDR2 as immunogens'. Is the major reason not that the CDR3 of the V γ 5 and V γ 6 chain of the invariant V γ 5V δ 1 and V γ 6V δ 1 $\gamma\delta$ cells are the same?

Reply: We agree that CDR3, the majority of binding contacts of the V γ 5 and V γ 6 chain of the invariant V γ 5V δ 1 and V γ 6V δ 1 are the same (ACWD). So, we selected peptides from CDR1 or CDR2 as immunogens We changed as follows

“The most likely immunogenic epitopes for staining the V γ 6 chain lie within the hypervariable CDR that provides the majority of binding contacts. However, CDR3 of the V γ 5 and V γ 6 chain of the invariant V γ 5V δ 1 and V γ 6V δ 1 are the same. So, we selected peptides from CDR1 or CDR2 as immunogens (WHO-IUIS, 1995).”

(Line 80–83, page 5–6 in revised manuscript)

Spelling:

Line 13: were located

Line 44: 'to' should be 'no'?

Line 83: titers

Reply: We corrected them.

April 26, 2019

RE: Life Science Alliance Manuscript #LSA-2019-00363-TR

Dr. Shinya Hatano
Division of Host Defense
Medical Institute of Bioregulation
3-1-1 Maidashi, Higashi-ku
Fukuoka, Fukuoka 812-8582
Japan

Dear Dr. Hatano,

Thank you for submitting your revised manuscript entitled "Development of a new monoclonal antibody specific to mouse Vy6 chain". We appreciate your response to the concerns previously raised by the reviewers and the introduced changes, and we would thus be happy to publish your paper in Life Science Alliance pending final small revisions:

- I would like to suggest some text edits, please see the file attached
- please add a scale bar to the H&E stainings in Fig 5D

A. FINAL FILES:

B. MANUSCRIPT ORGANIZATION AND FORMATTING:

Sincerely,

April 29, 2019

RE: Life Science Alliance Manuscript #LSA-2019-00363-TRR

Dr. Shinya Hatano
Division of Host Defense
Medical Institute of Bioregulation
3-1-1 Maidashi, Higashi-ku
Fukuoka, Fukuoka 812-8582
Japan

Dear Dr. Hatano,

Thank you for submitting your Resource entitled "Development of a new monoclonal antibody specific to mouse Vy6 chain". It is a pleasure to let you know that your manuscript is now accepted for publication in Life Science Alliance. Congratulations on this interesting work.

DISTRIBUTION OF MATERIALS:

Again, congratulations on a very nice paper. I hope you found the review process to be constructive and are pleased with how the manuscript was handled editorially. We look forward to future exciting submissions from your lab.

Sincerely,
